# The Effect of Network Width on the Performance of Large-batch Training

**Lingjiao Chen**[1] , **Hongyi Wang**[1] , **Jinman Zhao**[1],
**Paraschos Koutris,** [1] **Dimitris Papailiopoulos**[2]
[1]Department of Computer Sciences,    [2]Department of Electrical and Computer Engineering
University of Wisconsin-Madison

## Abstract

Distributed implementations of mini-batch stochastic gradient descent (SGD) suffer from communication overheads, attributed to the high frequency of gradient updates inherent in small-batch training. Training with large batches can reduce these overheads; however it besets the convergence of the algorithm and the generalization performance. In this work, we take a first step towards analyzing how the structure (width and depth) of a neural network affects the performance of large-batch training. We present new theoretical results which suggest that–for a fixed number of parameters–wider networks are more amenable to fast large-batch training compared to deeper ones. We provide extensive experiments on residual and fully-connected neural networks which suggest that wider networks can be trained using larger batches without incurring a convergence slow-down, unlike their deeper variants.

## 1   Introduction

Distributed implementations of stochastic optimization algorithms have become the standard in large-scale model training [1, 2, 3, 4, 5, 6, 7]. Most machine learning frameworks, including Tensorflow [1], MxNet [4], and Caffe2 [7], implement variants of mini-batch SGD as their default distributed training algorithm. During a distributed iteration of mini-batch SGD a *parameter server* (PS) stores the global model, and $P$ *compute nodes* evaluate a total of $B$ gradients; $B$ is commonly referred to as the *batch size*. Once the PS receives the sum of these $B$ gradients from every compute node, it applies them to the global model and sends the model back to the compute nodes, where a new distributed iteration begins.

The main premise of a distributed implementation is speedup gains, *i.*e., how much faster training takes on $P$ vs 1 compute node. In practice, these gains usually saturate beyond a few 10s of compute nodes [6, 8, 9]. This is because communication becomes the bottleneck, *i.*e., for a fixed batch of $B$ examples, as the number of compute nodes increases, these nodes will eventually spend more time communicating gradients to the PS rather than computing them. To mitigate this bottleneck, a plethora of recent work has studied low-precision training and gradient sparsification, *e.*g., [10, 11, 12].

An alternative approach to alleviate these overheads is to increase the batch size $B$, since $B$ directly controls the communication-computation ratio. Recent work develops sophisticated methods that enable large-batch training on state-of-the-art models and data sets [13, 14, 15]. At the same time, several studies suggest that large-batch training can affect the generalizability of the models [16], can slow down convergence [17, 18, 19], and is more sensitive to hyperparameter mis-tuning [20].

Several theoretical results [21, 18, 22, 19, 17] suggest that, when the batch size $B$ becomes larger than a problem-dependent threshold $B^*$, the total number of iterations to converge significantly

increases, rendering the use of larger $B$ a less viable option. Some of these studies, implicitly or explicitly, indicate that the threshold $B^*$ is controlled by the similarity of the gradients in the batch.

In particular, [19] shows that the measure of *gradient diversity* directly controls the relationship of $B$ and the convergence speed of mini-batch SGD. Gradient diversity measures the similarity of concurrently processed gradients, and [19] shows theoretically and experimentally that the higher the diversity, the more amenable a problem is to fast large-batch training, and by extent to speedup gains in a distributed setting.

A large volume of work has focused on how the structure of neural networks can affect the complexity or capacity [23, 24, 25] of the model, its representation efficiency [26], and its prediction accuracy [27, 28]. However, there is little work towards understanding how the structure of a neural network affects its amenability to distributed speedup gains.

In this work, through analyzing the gradient diversity of different network architectures, we take a step towards addressing the following question: *How does the structure of a neural network affect its amenability to fast large-batch training?*

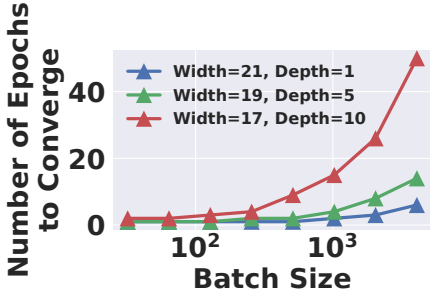

Figure 1: Impact of neural network structure on amenability to large-batch training. This is for fully-connected models with ReLUs on M-NIST. For each fully-connected network, we vary the batch size and measure the number of epochs to converge to $96\%$ accuracy on M-NIST. Wider and shallower networks require less epochs to converge than narrower and deeper ones, which suggests that the former are more suitable to scale out to more compute nodes.

**Our contribution** We establish a theoretical connection between the structure (depth and width) of neural networks and their gradient diversity, which is an indicator of how large batch size can become, without slowing down the speed of convergence [19]. In particular, we prove how gradient diversity varies as a function of width and depth for two types of networks: 2-layer fully-connected linear and non-linear neural networks, and multi-layer fully-connected linear neural networks. Our theoretical analysis indicates that, perhaps surprisingly, gradient diversity increases monotonically as width increases and depth decreases. On a high-level, wider networks provide more space for the gradients to become diverse. This result suggests that wider and shallower networks are more amenable to fast large-batch training compared to deeper ones. Figure 1 provides an illustrative example of this phenomenon.

We provide extensive experimental results that support our theoretical findings. We present experiments on fully-connected and residual networks on CIFAR10, MNIST, EMNIST, Gisette, and synthetic datasets. In our experimental setting, we fix the number of network parameters, vary the depth and width, and measure (after tuning the step size) how many passes over the data it takes to reach an accuracy of $\epsilon$ with batch size $B$. We observe that for all networks there exists a threshold $B^*$, and setting the batch size larger than the threshold leads to slower convergence. The observed threshold $B^*$ becomes smaller when the network becomes deeper, validating our theoretical result that deeper networks are less amenable to fast large-batch training.

To summarize the main message of our work, communication bottlenecks in distributed mini-batch SGD can be partially overcome not only by designing communication-efficient algorithms, but also by optimizing the architecture of the neural network at hand in order to enable large-batch training.

## 2 Related Work

**Mini-batch** The choice of an optimal batch size has been studied for non-strongly convex models [21], least square regression [22], and SVMs [29]. Other works propose methods that automatically choose the batch size on the fly [30, 31]. Mini-batch algorithms can be combined with accelerated gradient descent algorithms [32], or using dual coordinate descent [33, 34]. Mini-batch proximal algorithms are presented in [35]. While previous work mainly focuses on (strongly) convex models, or specific models (*e*.g., least square regression, SVMs), our work studies how neural network structure can affect the optimal batch size.

**Gradient Diversity** Previous work indicates that mini-batch can achieve better convergence rates by increasing the diversity of gradient batches, *e*.g., using stratified sampling [36], Determinantal

Point Processes [37], or active sampling [38]. The notion of similarity between gradients and how it affects convergence performance has been studied in several papers [17, 18, 19]. A formal definition and analysis of gradient diversity is given in [19], which establishes the connection between gradient diversity and maximum batch size for convex and nonconvex models. To the best of our knowledge, none of the existing works relates gradient diversity (and thus the optimal batch size) with the structure of a neural network.

**Width vs Depth in Artificial Neural Networks** There has been an increasing interest and debate on the qualities of deep versus wide neural networks. [23] suggests that deep networks have larger complexity than wide networks and thus may be able to obtain better models. [26] proves that deep networks can approximate sum products more efficiently than wide networks. Meanwhile, [39] shows that a class of wide ResNets can achieve at least as high accuracy as deep ResNets. [40] presents two classes of networks, one shallow and one deep, that achieve similar prediction error for saliency prediction. In fact, [41] shows that well-designed shallow neural networks can outperform many deep neural networks. More recently, [27] shows that using a dense structure, wider yet shallower networks can significantly improve the accuracy compared to deeper networks. In addition, [42] shows that larger widths leads to better optimization landscape. While previous work has mainly studied the effect of network structure on prediction accuracy, we focus on its effect on the optimal choice of batch size for distributed computation.

## 3 Setup and Preliminaries

In this section, we present the necessary background and problem setup.

**Mini-batch SGD** The process of training a model from data can be cast as an optimization problem known as *empirical risk minimization* (ERM):

$$\min_{\mathbf{w}} \frac{1}{n} \sum_{i=1}^{n} \ell(\mathbf{w}; (\mathbf{x}_i, y_i))$$

where $\mathbf{x}_i \in \mathbb{R}^m$ represents the $i$th data point, $n$ is the total number of data points, $\mathbf{w} \in \mathbb{R}^d$ is a parameter vector or model, and $\ell(\cdot; \cdot)$ is a loss function that measures the prediction accuracy of the model on each data point. One way to approximately solve the above ERM is through mini-batch stochastic gradient descent (SGD), which operates as follows:

$$\mathbf{w}_{(k+1)B} = \mathbf{w}_{kB} - \gamma \sum_{\ell=kB}^{(k+1)B-1} \nabla f_{s_\ell}(\mathbf{w}_{kB}), \tag{3.1}$$

where each index $s_\ell$ is drawn uniformly at random from $[n]$ with replacement. We use $\mathbf{w}$ with subscript $kB$ to denote the model we obtain after $k$ distributed iterations, *i.e.*, a total number of $kB$ gradient updates. In related studies there is often a normalization factor included in the batch computation, but here we subsume that in the step size $\gamma$.

**Gradient diversity and speed of convergence** *Gradient diversity* measures the degree to which individual gradients of the loss function are different from each other.

**Definition 1** (Gradient Diversity [19]). *We refer to the following ratio as gradient diversity*

$$\Delta_{\mathcal{S}}(\mathbf{w}) := \frac{\sum_{i=1}^{n} \|\nabla f_i(\mathbf{w})\|_2^2}{\|\sum_{i=1}^{n} \nabla f_i(\mathbf{w})\|_2^2} = \frac{\sum_{i=1}^{n} \|\nabla f_i(\mathbf{w})\|_2^2}{\sum_{i=1}^{n} \|\nabla f_i(\mathbf{w})\|_2^2 + \sum_{i \neq j} \langle \nabla f_i(\mathbf{w}), \nabla f_j(\mathbf{w}) \rangle}.$$

The gradient diversity $\Delta_{\mathcal{S}}(\mathbf{w})$ is large when the inner products between the gradients taken with respect to different data points are small. Equipped with the notion of gradient diversity, we define a batch size bound $B_{\mathcal{S}}(\mathbf{w})$ for each data set $\mathcal{S}$ and each $\mathbf{w}$ as follows:

$$B_{\mathcal{S}}(\mathbf{w}) := n \cdot \Delta_{\mathcal{S}}(\mathbf{w}).$$

The following result [19] uses the notion of gradient diversity to capture the convergence rate of mini-batch SGD.

**Lemma 1.** *[Theorem 3 in [19],Informal] Suppose $B \leq \delta \cdot n\Delta_{\mathcal{S}}(\mathbf{w}) + 1, \forall \mathbf{w}$ in each iteration. If serial SGD achieves an $\epsilon$-suboptimal solution after $T$ gradient updates, then using the same step-size as serial SGD, mini-batch SGD with batch-size $B$ can achieve a $(1 + \frac{\delta}{2})\epsilon$-suboptimal solution after the same number of gradient updates/data pass ( i.e., $T/B$ iterations).*

The above result is true for both convex and non-convex problems, and its main message is that mini-batch SGD does not suffer from speedup saturation as long as the batch size is smaller than $n \cdot \Delta_{\mathcal{S}}(\mathbf{w})$ (up to a constant factor). Moreover, [19] also shows that this is a worst-case optimal bound, *i.e.*, (roughly) if the batch size is larger than $n$ times the gradient diversity, there exists some model such that the convergence rate of mini-batch SGD is slower than that of serial SGD.

The main theoretical question that we study in this work is the following: *how does gradient diversity change as neural networks' structure (depth and width) varies?*

**Fully-connected Neural Networks**    We consider both linear and non-linear fully connected networks, with $L \geq 2$ layers. We denote by $K_\ell$ the *width* (number of nodes) of the $\ell$-th layer, where $\ell \in \{0, \dots, L\}$. The first layer corresponds to the input of dimension $d$, hence $K_0 = d$. The last layer corresponds to the single output of the neural network, hence $K_L = 1$. The weights of the edges that connect the $\ell$ and $\ell - 1$ layers, where $l \in \{1, \dots, L\}$, are represented by the matrix $W_\ell \in \mathbb{R}^{K_\ell \times K_{\ell-1}}$. For the sake of simplicity, we will express the collection of weights (*i.e.*, the model) as $\mathbf{w} = (W_1, W_2, \dots, W_L)$.

A general neural network (NN) with $L \geq 2$ layers can be described as a collection of matrices $W_1, \dots, W_L$, where $W_\ell \in \mathbb{R}^{K_\ell \times K_{\ell-1}}$, together with a (generally nonlinear) *activation function* $\sigma(\cdot)$. The output of a NN (or LNN) on input data point $\mathbf{x}_i$ is then defined as $\hat{y}_i = W_L \cdot \sigma(\cdots \sigma(W_2 \cdot \sigma(W_1 \cdot \mathbf{x}_i)))$. There are different types of activation that we study,*i.e.*, $\tanh(x)$, the *softsign* function $\frac{x}{1+|x|}$, $\arctan(x)$, and the ReLU function $\max\{0, x\}$. For linear neural networks (LNNs), we denote $W = \prod_{\ell=1}^{L} W_\ell = W_L \cdot W_{L-1} \cdots W_1$. We will also write $W_{\ell,p,q}$ to denote the element in the $p$-th row and $q$-th column of matrix $W_\ell$.

The output of the neural network with input $\mathbf{x}_i$ is defined as $\hat{y}_i$. Throughout the theory part of this paper, we will use the *square loss function* to measure the error, which we denote for the $i$-th data point as $f_i = \frac{1}{2}(\hat{y}_i - y_i)^2$. Further, we assume that the data is achievable, i.e., there exists $W^*_{\ell,p,q}$ such that the loss function is 0 on each data point when $W_{\ell,p,q} = W^*_{\ell,p,q}$.

# 4   Main Results

In this section, we present a theoretical analysis on how structural properties of a neural network, and in particular the *depth* and *width*, influence the gradient diversity, and hence the convergence rate of mini-batch SGD for varying batch size $B$. All proofs are left to the Appendix.

In the following derivations, we will assume that the labels $\{y_1, \dots, y_n\}$ of the $n$ data points are *realizable*, *i.e.*, there exist a network of $L$ layers that on input $x_i$ outputs $y_i$. Our results are presented as probabilistic statements, and for almost all weight matrices.

**Warmup: 2-Layer Linear Neural Networks**    Our first result concerns the case of a simple 2-layer linear neural network with one hidden layer. To simplify notation, we will denote the width of the hidden layer with $K = K_1$. Further, $\Theta(\cdot)$ and $\Omega(\cdot)$ are used in their standard meaning. The main result can be stated as follows:

**Theorem 1.** *Consider a 2 LNN. Let the weights $W_{l,p,q}, W^*_{l,p,q}$ for $l \in \{1, 2\}$ and $\mathbf{x}_i$ be independently drawn random variables, such that their $k$-th order moments for $k \leq 4$ are bounded in a postive interval. Then, with arbitrary constant probability, the following holds:*

$$B_{\mathcal{S}}(\mathbf{w}) \geq \frac{\Theta(nKd)}{\Theta(Kn + dn + Kd)}$$

For sufficiently large $n$, the above ratio on the batch size scales like $\frac{\Theta(Kd)}{\Theta(K+d)}$. This ratio is always increasing as a function of the *width* of the hidden layer, which implies that larger width allows for a larger batch size.

**2-Layer Nonlinear Neural Networks**    As a next step in our theoretical analysis, we analyze general 2-layer NNs with a nonlinear activation function $\sigma$.

**Theorem 2.** *Consider a 2-layer NN with a monotone activation function $\sigma$ such that for every $x$ we have: $-\sigma(x) = \sigma(-x)$, and both $|\sigma(x)|$ and $\sup_x\{x\sigma'(x)\}$ are bounded. Let the weights*

$W_{l,p,q}, W^*_{l,p,q}$ *for* $l \in \{1, 2\}$ *and* $\mathbf{x}_i$ *be i.i.d. random variables from* $\mathcal{N}(0, 1)$*. Then, with high probability, the following holds:*

$$\frac{\mathbb{E}[n \sum_{i=1}^{n} ||\nabla f_i||_2^2]}{\mathbb{E}[|| \sum_{i=1}^{n} \nabla f_i||_2^2]} \geq \Omega(\frac{Kd^2}{Kd + K + d}).$$

*where the expectation is over* $W_2, W_2^*$*.*

We should remark here that the above bound is weaker than the one obtained for the case of 2-layer LNNs, since it bounds the ratio of the expectations, and not the expectation of the ratio (the batch size bound). Nevertheless, we conjecture that the batch size bound concentrates, and thus the above theorem can approximate the batch size bound well.

Another remark is that several commonly used activation functions in NNs, such as $\tanh$, $\arctan$, and the softsign function satisfy the assumptions of the above theorem. The same trends can be observed here as in the case of 2-layer LNNs: $(i)$ larger width leads to a larger gradient diversity, and thus faster convergence of distributed mini-batch SGD, and $(ii)$ the ratio can never exceed $\Omega(d)$.

**Multilayer Linear Neural Networks**   We generalize here our result for 2-layer LNNs to general multilayer LNNs of arbitrary depth $L \geq 2$. Below is our main result.

**Theorem 3.** *Let the weight values* $W_{l,p,q}$ *for* $l \in \{1, \ldots, L\}$ *and* $\mathbf{x}_i$ *be independently drawn random variables from* $\mathcal{N}(0, 1)$*. Consider a multilayer LNN where* $f_i = \frac{1}{2}(W\mathbf{x}_i - W^*\mathbf{x}_i)^2 = \frac{1}{2}(\prod_{\ell=1}^{L} W_\ell \mathbf{x}_i - \prod_{\ell=1}^{L} W_\ell^* \mathbf{x}_i)^2$*. Assuming that* $K_\ell \geq 2$ *for every* $\ell \in \{0, \ldots, L-1\}$*, and that* $n$ *is sufficiently large, then we have:*

$$\rho = \frac{\mathbb{E}[n \sum_{i=1}^{n} ||\nabla f_i||_2^2]}{\mathbb{E}[|| \sum_{i=1}^{n} \nabla f_i||_2^2]} \geq \frac{L}{\sum_{\phi=1}^{L-1} \frac{L-\phi}{K_\phi - 1} + \frac{2L}{d-1}}. \tag{4.1}$$

Again, note that the above bound is weaker than the one obtained for the case of 2-layer LNNs, since it bounds the ratio of the expectations, and not the expectation of the ratio. It is believed that the denominator and numerator should concentrate around their expectations (as was the case in Theorem 1) and thus ratio of the expectation reflects the expectation of the ratio. Whether this can be proved remains an interesting open question.

We next discuss the implications of Theorem 3 on the convergence rate of mini-batch SGD. To analyze the behavior of the bound, consider the simple case where all the hidden layers ($l = 1, \ldots, L-1$) have exactly the same width $K$. In this case, the ratio in Eq. (4.1) becomes:

$$\rho \geq \frac{1}{\frac{L-1}{2(K-1)} + \frac{2}{d-1}} = \Theta\left(\frac{dK}{dL + K}\right)$$

There are three takeaways from the above bound. First, by increasing the width $K$ of the LNN, the ratio increases as well, which implies that the convergence rate increases. Second, the effect of the depth $L$ is the opposite: by increasing the depth, the ratio decreases. Third, the ratio can never exceed $\Theta(d)$, but it can be arbitrarily small. Suppose now that we fix the total number of weights in the LNN, and then start increasing the width of each layer (which means that the depth will decrease). In this case, the ratio will also increase.

We conclude by noting that the same behavior of the bound w.r.t. width and depth can be observed if we drop the simplifying assumption that all layers have the same width.

## 5   Experiments

In this section, we provide empirical results on how the structure of a neural network (width and depth) impacts its amenability to large-batch training using various datasets and network architectures. Our main findings are three-fold:

1. For all neural networks we used, there exists a threshold $B^*$, such that using batch size larger than this threshold induces slower convergence;
2. The threshold of wider neural networks is often larger than that of deeper ones;

| Dataset | Synthetic | MNIST | Cifar10 | EMNIST | Gisette |
|---|---|---|---|---|---|
| # datapoints | 10,000 | 70,000 | 60,000 | 131,600 | 6,000 |
| Model | linear FC | FC/LeNet | ResNet-18/34 | FC | FC |
| # Classes | $+\infty$ | 10 | 10 | 47 | 2 |
| # Parameters | 16k | 16k / 431k | 11m / 21m | 16k | 262k |
| Converged Accuracy | $10^{-12}$ (loss) | 96% / 98% | 95% | 65% | 95% |

Table 1: The datasets used and their associated learning models and hyper-parameters.

3. When using the same large batch size, almost all wider neural networks need much fewer epochs to converge compared to their deeper counterparts.

Those findings validate our theoretical analysis and suggest that wider neural networks are indeed more amenable to large-batch training and thus more suitable to scale out.

**Implementation and Setup** We implemented our experimental pipeline in Keras [43], and conducted all experiments on p2.xlarge instances on Amazon EC2. All results reported are averaged from 5 independent runs.

**Datasets and Networks** Table 1 summarizes the datasets and networks used in the experiments. In the synthetic dataset, all data points were independently drawn from $\mathcal{N}(0, 1)$ as described by our theory results. A deep linear fully connected neural network (FC) whose weights were generated from $\mathcal{N}(0, 1)$ independently was used to produce the true labels. The task on the synthetic data is a regression task. We train linear FCs on the synthetic dataset. The real-world datasets we used include MNIST [44], EMNIST[45], Gisette [46], and CIFAR-10 [47], with appropriate networks ranging from linear, to non-linear fully connected ones, and to LeNet [48] and ResNet [28].

For each network, we fix the total number of parameters and vary its depth/number of layers $L$ and width $K$. For fully connected networks and LeNet, we vary depth $L$ from 1 to 10 and change $K$ accordingly to ensure the total number of parameters are approximately fixed. More precisely, we fix the total number of parameters $p$, and solve the following equations

$$d_{\text{in}} \times K + (L - 1) \times K^2 + K \times d_{\text{out}} = p.$$

where $d_{in}$ is the dimension of the data and $d_{out}$ is the size of output. For ResNet, we vary two parameters separately. We first vary the width and depth of the fully connected layers without changing the residual blocks. Next we fix the fully connected layers and change the number of blocks and convolution filters in each chunk. We refer to the building block in a residual function described in [28] as chunk. For ResNet-18/34 architecture, we use $[s_1, s_2, s_3, s_4]$ to denote a particular structure, where $s_1$ represents the number of blocks stacked in the first chunk, $s_2$ is the number of blocks stacked in the second chunk, etc. For varying depths, we incrementally increase or decrease one block in each chunk and adjust the number of convolutional filters in each block to meet the fixed number of parameters requirement.

For each combination of depth and width of a NN architecture, we train the model by setting a constant threshold on training accuracy for classification tasks, or loss for regression tasks. We then train the NN for a variety of batch sizes, in range of $2^i$, for $i \in \{5, \cdots, 12\}$. We tune the step size in the following way: (i) for all learning rates $\eta$ from a grid of candidate values, we run the training process with $\eta$ for 2 passes over the data; and then (ii) we choose $\hat{\eta}$ which leads to the lowest training loss after two epochs. An epoch represents a full pass over the data.

**Experimental Results** We first verify whether gradient diversity reflects the amenability to large batch training. For each linear FC network with fixed width and depth, we measure its gradient diversity every ten epochs and compute the average. Figure 2(a) shows how the averaged gradient diversity varies as depth/width changes, while Figure 2(b) presents the largest batch to converge for each network within a pre-set number of epochs. Both of them increase as the width $K$ of the networks increases. In fact, as shown in Figure 2(c), the largest batch size that does not impact the convergence rate grows monotonically w.r.t the gradient diversity. This validates our theoretical analysis that gradient diversity can be used to capture the amenability to large batch training.

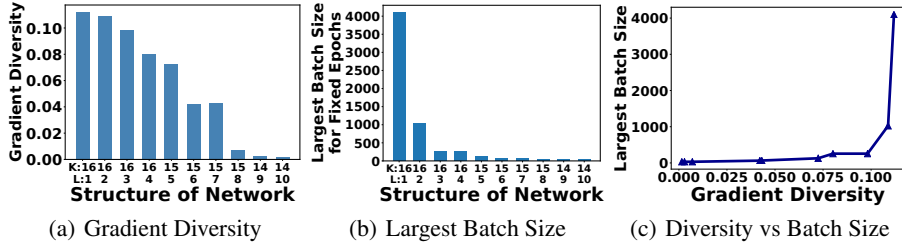

(a) Gradient Diversity     (b) Largest Batch Size     (c) Diversity vs Batch Size

Figure 2: The effect of gradient diversity for linear FCs trained on the synthetic dataset for a regression task. (a) Gradient diversity for different width/depth (b) Largest batch size to converge to loss $10^{-12}$, within a pre-set number (*i.e.*, 14) of epochs. (c) Largest batch size v.s. gradient diversity.

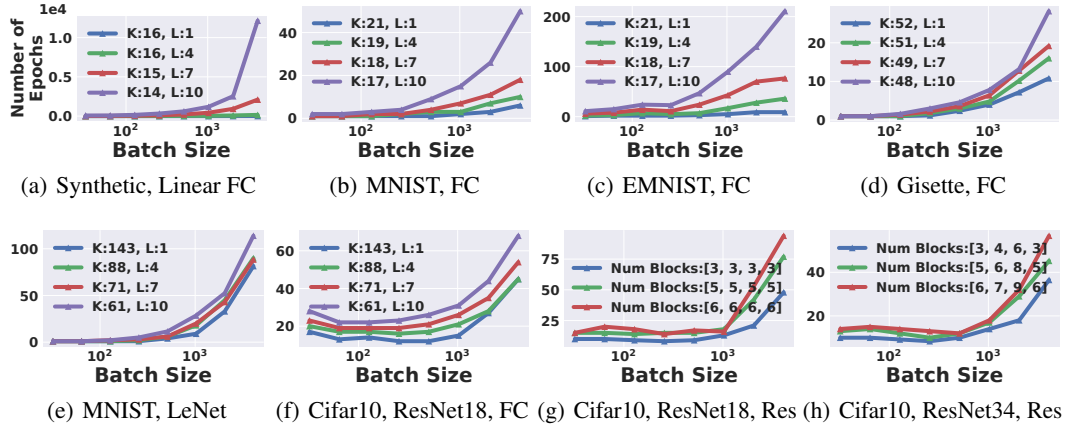

(a) Synthetic, Linear FC    (b) MNIST, FC    (c) EMNIST, FC    (d) Gisette, FC

(e) MNIST, LeNet    (f) Cifar10, ResNet18, FC    (g) Cifar10, ResNet18, Res (h) Cifar10, ResNet34, Res

Figure 3: Number of epochs needed to converge to the same loss / accuracy given in Table 1. $K$ represents width, and $L$ depth. In (f) We fix the residual blocks of ResNet 18 and only vary the fully-connected parts. In (g) and (h), we fix the fully connected layers and vary the residual blocks of ResNet 18 and ResNet 34.

Next, we study the number of epochs needed to converge when different batch sizes are used for real-world datasets. First, for almost all network architectures, there exists a batch size threshold, such that using a batch size larger than this, requires more epochs for convergence, consistent with the observations in [19]. For example, in Figure 3(b), when the batch size is smaller than 256, the FC network with width $K = 17$ and depth $L = 10$ needs a small number (2 to 3) of epochs to converge. But when the batch size becomes larger than 256, the number of epochs necessary for convergence increases significantly, *e.g.*, it takes 50 epochs to converge when batch size is 4096. Moreover, we observe that the threshold increases as width increases. Again as shown in Figure 3(b), the batch-size threshold for the FC network with $L = 10$ is 256, but goes up to 1024 with $L = 1$. Furthermore, when using the same large batch size, wider networks tend to require fewer epochs to converge than the deeper ones. In Figure 3(c), for instance, using the same batch size of 4096, the required epochs to converge decreases from 211 to 9 as width $K$ increases from 17 to 21. Those trends are similar for all FC networks we used in the experiments.

When it comes to ResNets and LeNet, the trends are not always as sharp. This is expected since our theoretical analysis does not cover such cases, but the main trend can still be observed. For example, as shown in Figures 3(e) and 3(f), for a fixed batch size, increasing the width almost always leads to a decrease in number of epochs for convergence. Figure 4, depicts the exact number of epochs to converge for each network architecture, and plots them as a heatmap. It is interesting to see that for ResNet, there is a small fraction of cases where increase of depth can also reduce the number of epochs for convergence.

In many practical applications, only a reasonable and limited number of data passes is performed due to time and resources constraints. Thus, we also study how the structure of a network affects the largest possible batch size to converge within a fixed number of epochs/data passes to a pre-specified accuracy. As shown in Figure 5, neural networks with larger width $K$ usually allow much larger batch sizes to converge within a small, pre-set number of total epochs. This is especially beneficial in

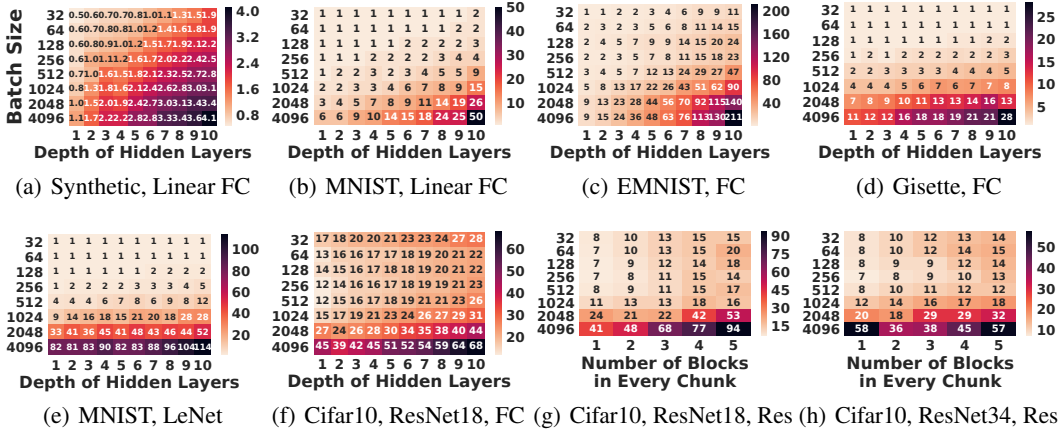

Figure 4: Heatmap on number of epochs needed to converge to loss / accuracy defined in Table 1. We report the $\log_{10}$ of the epochs for (a) and the real epochs for the others.

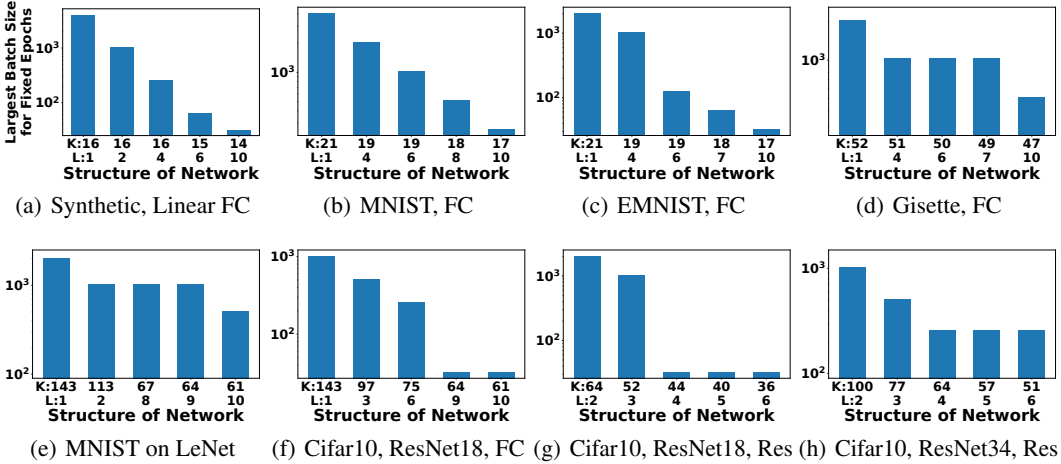

Figure 5: Largest possible batch size to converge within a fixed number of epochs.

the scenarios of large-scale distributed learning, since increasing the batch size can result in more speedup gains due to a reduction in the total amount of communication. Finally, we should note that the largest batch size differs among different networks, as well as different datasets. This is because gradient diversity is both data-dependent and model-dependent.

## 6 Conclusion

In this paper, we study how the structure of a neural network affects the performance of large-batch training. Through the lens of gradient diversity, we quantitatively connect a network's amenability to larger batches during training with its depth and width. Extensive experimental results, along with theoretical analysis, demonstrate that for a large class of neural networks, increasing width leads to larger gradient diversity and thus allows for a larger batch training that is always beneficial for distributed computation.

In the future, we plan to explore how a particular structure, *e*.g., convolutional filters, residual blocks, etc, affects gradient diversity. From a practical perspective, we argue that it is important to consider the architecture of a network with regards to its amenability for speedups in a distributed setting. Hence, we plan to explore how one can fine-tune a network so that large-batch training is enabled, and communication bottlenecks are minimized. Another direction is to quantitatively study how the generalization error is affected.

