[Supplementary Material]

## Acknowledgement

This work was supported in part by MADLab at University of Wisconsin - Madison and AWS Cloud Credits for Research from Amazon.

## References

[1] Martín Abadi, Paul Barham, Jianmin Chen, Zhifeng Chen, Andy Davis, Jeffrey Dean, Matthieu Devin, Sanjay Ghemawat, Geoffrey Irving, Michael Isard, Manjunath Kudlur, Josh Levenberg, Rajat Monga, Sherry Moore, Derek Gordon Murray, Benoit Steiner, Paul A. Tucker, Vijay Vasudevan, Pete Warden, Martin Wicke, Yuan Yu, and Xiaoqiang Zheng. Tensorflow: A system for large-scale machine learning. In *OSDI 2016*, pages 265–283, 2016.

[2] Mu Li, David G. Andersen, Jun Woo Park, Alexander J. Smola, Amr Ahmed, Vanja Josifovski, James Long, Eugene J. Shekita, and Bor-Yiing Su. Scaling distributed machine learning with the parameter server. In *OSDI 2014*, pages 583–598, 2014.

[3] John C. Duchi, Alekh Agarwal, and Martin J. Wainwright. Distributed dual averaging in networks. In *NIPS 2010*, pages 550–558, 2010.

[4] Tianqi Chen, Mu Li, Yutian Li, Min Lin, Naiyan Wang, Minjie Wang, Tianjun Xiao, Bing Xu, Chiyuan Zhang, and Zheng Zhang. Mxnet: A flexible and efficient machine learning library for heterogeneous distributed systems. *CoRR*, abs/1512.01274, 2015.

[5] Jianmin Chen, Rajat Monga, Samy Bengio, and Rafal Józefowicz. Revisiting distributed synchronous SGD. *CoRR*, abs/1604.00981, 2016.

[6] Jeffrey Dean, Greg Corrado, Rajat Monga, Kai Chen, Matthieu Devin, Quoc V. Le, Mark Z. Mao, Marc'Aurelio Ranzato, Andrew W. Senior, Paul A. Tucker, Ke Yang, and Andrew Y. Ng. Large scale distributed deep networks. In *NIPS 2012*, pages 1232–1240, 2012.

[7] Caffe2: A new lightweight, modular, and scalable deep learning framework.

[8] Hang Qi, Evan R. Sparks, and Ameet Talwalkar. Paleo: A performance model for deep neural networks. In *ICLR*, 2017.

[9] Hanlin Tang, Xiangru Lian, Ming Yan, Ce Zhang, and Ji Liu. D$_2$: Decentralized training over decentralized data. *CoRR*, abs/1803.07068, 2018.

[10] Dan Alistarh, Demjan Grubic, Jerry Li, Ryota Tomioka, and Milan Vojnovic. QSGD: communication-efficient SGD via gradient quantization and encoding. In *NIPS 2017*, pages 1707–1718, 2017.

[11] Wei Wen, Cong Xu, Feng Yan, Chunpeng Wu, Yandan Wang, Yiran Chen, and Hai Li. Terngrad: Ternary gradients to reduce communication in distributed deep learning. In *NIPS 2017*, pages 1508–1518, 2017.

[12] Yujun Lin, Song Han, Huizi Mao, Yu Wang, and William J. Dally. Deep gradient compression: Reducing the communication bandwidth for distributed training. *CoRR*, abs/1712.01887, 2017.

[13] Priya Goyal, Piotr Dollár, Ross B. Girshick, Pieter Noordhuis, Lukasz Wesolowski, Aapo Kyrola, Andrew Tulloch, Yangqing Jia, and Kaiming He. Accurate, large minibatch SGD: training imagenet in 1 hour. *CoRR*, abs/1706.02677, 2017.

[14] Elad Hoffer, Itay Hubara, and Daniel Soudry. Train longer, generalize better: closing the generalization gap in large batch training of neural networks. In *NIPS 2017*, pages 1729–1739, 2017.

[15] Yang You, Igor Gitman, and Boris Ginsburg. Scaling SGD batch size to 32k for imagenet training. *CoRR*, abs/1708.03888, 2017.

[16] Nitish Shirish Keskar, Dheevatsa Mudigere, Jorge Nocedal, Mikhail Smelyanskiy, and Ping Tak Peter Tang. On large-batch training for deep learning: Generalization gap and sharp minima. *CoRR*, abs/1609.04836, 2016.

[17] Prateek Jain, Sham M. Kakade, Rahul Kidambi, Praneeth Netrapalli, and Aaron Sidford. Parallelizing stochastic gradient descent for least squares regression: mini-batching, averaging, and model misspecification. *CoRR*, abs/1610.03774, 2018.

[18] Siyuan Ma, Raef Bassily, and Mikhail Belkin. The power of interpolation: Understanding the effectiveness of SGD in modern over-parametrized learning. *CoRR*, abs/1712.06559, 2017.

[19] Dong Yin, Ashwin Pananjady, Maximilian Lam, Dimitris S. Papailiopoulos, Kannan Ramchandran, and Peter Bartlett. Gradient diversity: a key ingredient for scalable distributed learning. In *AISTATS 2018*, pages 1998–2007, 2018.

[20] Dominic Masters and Carlo Luschi. Revisiting small batch training for deep neural networks. *CoRR*, abs/1804.07612, 2018.

[21] Ofer Dekel, Ran Gilad-Bachrach, Ohad Shamir, and Lin Xiao. Optimal distributed online prediction using mini-batches. *Journal of Machine Learning Research*, 13:165–202, 2012.

[22] Prateek Jain, Sham M. Kakade, Rahul Kidambi, Praneeth Netrapalli, and Aaron Sidford. Parallelizing stochastic approximation through mini-batching and tail-averaging. *CoRR*, abs/1610.03774, 2016.

[23] Monica Bianchini and Franco Scarselli. On the complexity of neural network classifiers: A comparison between shallow and deep architectures. *IEEE Trans. Neural Netw. Learning Syst.*, 25(8):1553–1565, 2014.

[24] Andrew R. Barron. Approximation and estimation bounds for artificial neural networks. In *COLT 1991*, pages 243–249, 1991.

[25] Zhou Lu, Hongming Pu, Feicheng Wang, Zhiqiang Hu, and Liwei Wang. The expressive power of neural networks: A view from the width. In *NIPS 2017*, pages 6232–6240, 2017.

[26] Olivier Delalleau and Yoshua Bengio. Shallow vs. deep sum-product networks. In *NIPS 2011*, pages 666–674, 2011.

[27] Gao Huang, Zhuang Liu, Laurens van der Maaten, and Kilian Q. Weinberger. Densely connected convolutional networks. In *CVPR 2017*, pages 2261–2269, 2017.

[28] Kaiming He, Xiangyu Zhang, Shaoqing Ren, and Jian Sun. Deep residual learning for image recognition. In *CVPR 2016*, pages 770–778, 2016.

[29] Martin Takác, Avleen Singh Bijral, Peter Richtárik, and Nati Srebro. Mini-batch primal and dual methods for svms. In *ICML 2013*, pages 1022–1030, 2013.

[30] Michael P. Friedlander and Mark W. Schmidt. Erratum: Hybrid deterministic-stochastic methods for data fitting. *SIAM J. Scientific Computing*, 35(4), 2013.

[31] Soham De, Abhay Kumar Yadav, David W. Jacobs, and Tom Goldstein. Big batch SGD: automated inference using adaptive batch sizes. *CoRR*, abs/1610.05792, 2016.

[32] Andrew Cotter, Ohad Shamir, Nati Srebro, and Karthik Sridharan. Better mini-batch algorithms via accelerated gradient methods. In *NIPS 2011*, pages 1647–1655, 2011.

[33] Shai Shalev-Shwartz and Tong Zhang. Accelerated mini-batch stochastic dual coordinate ascent. In *NIPS 2013*, pages 378–385, 2013.

[34] Martin Takác, Peter Richtárik, and Nathan Srebro. Distributed mini-batch SDCA. *CoRR*, abs/1507.08322, 2015.

[35] Jialei Wang, Weiran Wang, and Nathan Srebro. Memory and communication efficient distributed stochastic optimization with minibatch prox. In *COLT 2017*, pages 1882–1919, 2017.

[36] Peilin Zhao and Tong Zhang. Accelerating minibatch stochastic gradient descent using stratified sampling. *CoRR*, abs/1405.3080, 2014.

[37] Cheng Zhang, Hedvig Kjellström, and Stephan Mandt. Stochastic learning on imbalanced data: Determinantal point processes for mini-batch diversification. *CoRR*, abs/1705.00607, 2017.

[38] Cheng Zhang, Cengiz Öztireli, Stephan Mandt, and Giampiero Salvi. Active mini-batch sampling using repulsive point processes. *CoRR*, abs/1804.02772, 2018.

[39] Sergey Zagoruyko and Nikos Komodakis. Wide residual networks. In *BMVC 2016*, 2016.

[40] Junting Pan, Elisa Sayrol, Xavier Giró i Nieto, Kevin McGuinness, and Noel E. O'Connor. Shallow and deep convolutional networks for saliency prediction. In *CVPR 2016*, pages 598–606, 2016.

[41] Zifeng Wu, Chunhua Shen, and Anton van den Hengel. Wider or deeper: Revisiting the resnet model for visual recognition. *CoRR*, abs/1611.10080, 2016.

[42] Quynh Nguyen and Matthias Hein. Optimization landscape and expressivity of deep cnns. In *Proceedings of the 35th International Conference on Machine Learning, ICML 2018, Stockholmsmässan, Stockholm, Sweden, July 10-15, 2018*, pages 3727–3736, 2018.

[43] François Chollet. keras. `https://github.com/fchollet/keras`, 2015.

[44] Yann LeCun, Corinna Cortes, and CJ Burges. Mnist handwritten digit database. *AT&T Labs [Online]. Available: http://yann. lecun. com/exdb/mnist*, 2, 2010.

[45] Gregory Cohen, Saeed Afshar, Jonathan Tapson, and André van Schaik. Emnist: an extension of mnist to handwritten letters. *arXiv preprint arXiv:1702.05373*, 2017.

[46] Chih-Chung Chang and Chih-Jen Lin. Libsvm: a library for support vector machines. *ACM transactions on intelligent systems and technology (TIST)*, 2(3):27, 2011.

[47] Alex Krizhevsky. Learning multiple layers of features from tiny images. Technical report, 2009.

[48] Yann LeCun, Léon Bottou, Yoshua Bengio, and Patrick Haffner. Gradient-based learning applied to document recognition. *Proceedings of the IEEE*, 86(11):2278–2324, 1998.

[49] L. Isserlis. On a formula for the product-moment coefficient of any order of a normal frequency distribution in any number of variables. *Biometrika*, 12(1/2):134–139, 1918.

# A  Proofs for 2-Layer LNNs

We will start by proving the following proposition.

**Proposition A.1.** *Assume that all weight values and the data points are independent random variable. And Further assume that their k-th order moments are bounded when $k \leq 4$. Within a weight matrix, all entries have the same moment value. All data points also have the same moment value. Then for any pair $i, j \in \{1, \ldots, n\}$:*

$$E(\langle \nabla f_i, \nabla f_j \rangle) = \begin{cases} \Theta(K^2 d^2) & \text{if } i = j \\ \Theta(Kd(K+d)) & \text{if } i \neq j \end{cases}$$

To prove the above proposition, we first define some notations. Let $M_{i,j}$ be the $j$-th moment of the entries of $W_i$, and $W_{x,j}$ be the $j$-th moment of the entries of data point $x_i$.

Let us prove the two cases separately. We first consider the case where $i = j$. We can write the inner product as

$$E\left(\|\nabla f_i\|^2\right) = \sum_{p=1}^{K} \sum_{q=1}^{d} E(\|\frac{\partial f_i}{\partial W_{1,p,q}}\|^2) + \sum_{q=1}^{K} E(\|\frac{\partial f_i}{\partial W_{2,1,q}}\|^2)$$

We will show in Lemma 2 that the first expectation is $\Theta(Kd)$, and in Lemma 3 that the second expectation is $\Theta(Kd^2)$. Plugging these results into the above equation gives the desired result.

**Lemma 2.** $E(\|\frac{\partial f_i}{\partial W_{1,p,q}}\|^2) = \Theta(Kd)$.

*Proof.* Note that $\frac{\partial f_i}{\partial W_{1,p,q}} = (\hat{y}_i - y_i)W_{2,1,p}x_{i,q} = (W_2 W_1 - W_2^* W_1^*)x_i W_{2,1,p}x_{i,q}$. We have

$$E(\|\frac{\partial f_i}{\partial W_{1,p,q}}\|^2) = E\left((W_2 W_1 - W_2^* W_1^*)\, x_i W_{2,1,p}x_{i,q}\right)^2$$

$$= E\left((W_2 W_1)\, x_i W_{2,1,p}x_{i,q}\right)^2 + E\left((W_2^* W_1^*)\, x_i W_{2,1,p}x_{i,q}\right)^2$$

$$= E\left(\sum_{s=1}^{K} W_{2,1,s}^2 \left(W_{1,s,:}x_i\right)^2 W_{2,1,p}^2 x_{i,q}^2\right) + E\left(\sum_{s=1}^{K} {W_{2,1,s}^*}^2 \left(W_{1,s,:}^* x_i\right)^2 W_{2,1,p}^2 x_{i,q}^2\right)$$

$$= E\left(\sum_{s=1}^{K} W_{2,1,s}^2 \left(\sum_{t=1}^{d} W_{1,s,t}^2 x_{i,t}^2\right) W_{2,1,p}^2 x_{i,q}^2\right) + E\left(\sum_{s=1}^{K} {W_{2,1,s}^*}^2 \left(\sum_{t=1}^{d} {W_{1,s,t}^*}^2 x_{i,t}^2\right) W_{2,1,p}^2 x_{i,q}^2\right)$$

$$= M_{1,2}\left((K-1)M_{2,2}^2 + M_{2,4}\right)\left((d-1)M_{x,2}^2 + M_{x,4}\right) + M_{1^*,2}KM_{2^*,2}^2\left((d-1)M_{x,2}^2 + M_{x,4}\right)$$

This concludes the proof. $\qquad\square$

**Lemma 3.** $E(\|\frac{\partial f_i}{\partial W_{2,1,q}}\|^2) = \Theta(Kd^2)$.

*Proof.* Note that $\frac{\partial f_i}{\partial W_{2,1,q}} = (\hat{y}_i - y_i)W_{1,q,:}x_i = (W_2 W_1 - W_2^* W_1^*)x_i W_{1,q,:}x_i$. We have

$$E(\|\frac{\partial f_i}{\partial W_{2,1,q}}\|^2) = E\left((W_2 W_1 - W_2^* W_1^*)\, x_i W_{1,q,:}x_i\right)^2$$

$$= E\left((W_2 W_1)\, x_i W_{1,q,:}x_i\right)^2 + E\left((W_2^* W_1^*)\, x_i W_{1,q,:}x_i\right)^2$$

$$= E\left(\sum_{s=1}^{K} W_{2,1,s}^2 \left(W_{1,s,:}x_i\right)^2 \left(W_{1,q,:}x_i\right)^2\right) + E\left(\sum_{s=1}^{K} {W_{2,1,s}^*}^2 \left(W_{1,s,:}^* x_i\right)^2 \left(W_{1,q,:}x_i\right)^2\right).$$

For the first term, we have

$$E\left(\sum_{s=1}^{K} W_{2,1,s}^2 \left(W_{1,s,:}x_i\right)^2 \left(W_{1,q,:}x_i\right)^2\right) = E\left(\sum_{s=1}^{K} W_{2,1,s}^2 \left(\sum_{t=1}^{d} W_{1,s,t}x_{i,t}\right)^2 \left(\sum_{u=1}^{d} W_{1,q,u}x_{i,u}\right)^2\right)$$

$$= \sum_{s=1}^{K} M_{2,2} E\left(\left(\sum_{t=1}^{d} W_{1,s,t}x_{i,t}\right)^2 \left(\sum_{u=1}^{d} W_{1,q,u}x_{i,u}\right)^2\right)$$

We now distinguish two cases. If $s \neq q$,

$$E\left(\left(\sum_{t=1}^{d} W_{1,s,t}x_{i,t}\right)^2 \left(\sum_{u=1}^{d} W_{1,q,u}x_{i,u}\right)^2\right) = E\left(\left(\sum_{t=1}^{d} W_{1,s,t}^2 x_{i,t}^2\right)\left(\sum_{u=1}^{d} W_{1,q,u}^2 x_{i,u}^2\right)\right)$$

$$= dM_{1,2}^2 M_{x,4} + d(d-1)M_{1,2}^2 M_{x,2}^2 = dM_{1,2}^2\left(M_{x,4} + (d-1)M_{x,2}^2\right) = \Theta(d^2)$$

If $s = q$,

$$E\left(\left(\sum_{t=1}^{d} W_{1,s,t} x_{i,t}\right)^2 \left(\sum_{u=1}^{d} W_{1,q,u} x_{i,u}\right)^2\right)$$

$$= E\left(\left(\sum_{t=1}^{d} W_{1,s,t}^2 x_{i,t}^2\right)\left(\sum_{u=1}^{d} W_{1,q,u}^2 x_{i,u}^2\right)\right)$$

$$+ E\left(\left(\sum_{t=1,t\neq v}^{d} W_{1,s,t} x_{i,t} W_{1,s,v} x_{i,v}\right)\left(\sum_{u=1,u\neq w}^{d} W_{1,q,u} x_{i,u} W_{1,q,w} x_{i,w}\right)\right)$$

$$= E\left(\left(\sum_{t=1}^{d} W_{1,s,t}^2 x_{i,t}^2\right)\left(\sum_{u=1}^{d} W_{1,q,u}^2 x_{i,u}^2\right)\right) + E\left(\left(2\sum_{t=1,t\neq v}^{d} W_{1,s,t}^2 x_{i,t}^2 W_{1,s,v}^2 x_{i,v}^2\right)\right)$$

$$= dM_{1,2}^2 M_{x,4} + 3d(d-1)M_{1,2}^2 M_{x,2}^2 = \Theta(d^2)$$

Combining the two cases, we have

$$E\left(\sum_{s=1}^{K} W_{2,1,s}^2 \left(W_{1,s,:} x_i\right)^2 \left(W_{1,q,:} x_i\right)^2\right) = \Theta(d^2)$$

For the second term, we have

$$E\left(\sum_{s=1}^{K} W_{2^*,1,s}^2 \left(W_{1^*,s,:} x_i\right)^2 \left(W_{1,q,:} x_i\right)^2\right)$$

$$= E\left(\sum_{s=1}^{K} W_{2^*,1,s}^2 \left(\sum_{t=1}^{d} W_{1^*,s,t} x_{i,t}\right)^2 \left(\sum_{u=1}^{d} W_{1,q,u} x_{i,u}\right)^2\right)$$

$$= \sum_{s=1}^{K} M_{2^*,2} E\left(\left(\sum_{t=1}^{d} W_{1^*,s,t} x_{i,t}\right)^2 \left(\sum_{u=1}^{d} W_{1,q,u} x_{i,u}\right)^2\right)$$

$$= \sum_{s=1}^{K} M_{2^*,2} E\left(\left(\sum_{t=1}^{d} W_{1^*,s,t}^2 x_{i,t}^2\right)\left(\sum_{u=1}^{d} W_{1,q,u}^2 x_{i,u}^2\right)\right)$$

$$= \sum_{s=1}^{K} M_{2^*,2}\left(dM_{1^*,2} M_{1,2} M_{x,4} + d(d-1)M_{1^*,2} M_{1,2} M_{x,2}^2\right)$$

$$= \Theta(Kd^2).$$

Combing the first term and the second term, we obtain the desired result. $\square$

We next consider the case $i \neq j$. In this case, we can write the inner product as

$$E(\langle \nabla f_i, \nabla f_j \rangle) = \sum_{p=1}^{K}\sum_{q=1}^{d} E(\langle \frac{\partial f_i}{\partial W_{1,p,q}}, \frac{\partial f_j}{\partial W_{1,p,q}} \rangle) + \sum_{q=1}^{K} E(\langle \frac{\partial f_i}{\partial W_{2,1,q}}, \frac{\partial f_j}{\partial W_{2,1,q}} \rangle)$$

As before, we will show in Lemma 4 that the first expectation is $\Theta(K)$, and in Lemma 5 that the second expectation is $\Theta(d(K+d))$. Plugging these results into the above equation gives the desired result.

**Lemma 4.** *If* $i \neq j$, $E(\langle \frac{\partial f_i}{\partial W_{1,p,q}}, \frac{\partial f_j}{\partial W_{1,p,q}} \rangle) = \Theta(K)$.

*Proof.* Note that $\frac{\partial f_i}{\partial W_{1,p,q}} = (\hat{y}_i - y_i)W_{2,1,p}x_{i,q} = (W_2 W_1 - W_2^* W_1^*)x_i W_{2,1,p}x_{i,q}$. We have

$$
E(\langle \frac{\partial f_i}{\partial W_{1,p,q}}, \frac{\partial f_j}{\partial W_{1,p,q}} \rangle)
$$
$$
= E\left( (W_2 W_1 - W_2^* W_1^*)\, x_i W_{2,1,p}x_{i,q} \left( W_2 W_1 - W_2^* W_1^* \right) x_j W_{2,1,p}x_{j,q} \right)
$$
$$
= E\left( W_2 W_1 x_i W_{2,1,p}x_{i,q} W_2 W_1 x_j W_{2,1,p}x_{j,q} \right) + E\left( W_2^* W_1^* x_i W_{2,1,p}x_{i,q} W_2^* W_1^* x_j W_{2,1,p}x_{j,q} \right)
$$
$$
= E\left( \sum_{s=1}^{K} W_{2,1,s}^2 \left( W_{1,s,:}x_i W_{1,s,:}x_j \right) W_{2,1,p}^2 x_{i,q}x_{j,q} \right)
$$
$$
+ E\left( \sum_{s=1}^{K} W_{2^*,1,s}^2 \left( W_{1^*,s,:}x_i W_{1^*,s,:}x_j \right) W_{2,1,p}^2 x_{i,q}x_{j,q} \right)
$$
$$
= E\left( \sum_{s=1}^{K} W_{2,1,s}^2 \left( \sum_{t=1}^{d} W_{1,s,t}^2 x_{i,t}x_{j,t} \right) W_{2,1,p}^2 x_{i,q}x_{j,q} \right)
$$
$$
+ E\left( \sum_{s=1}^{K} {W_{2,1,s}^*}^2 \left( \sum_{t=1}^{d} {W_{1,s,t}^*}^2 x_{i,t}x_{j,t} \right) W_{2,1,p}^2 x_{i,q}x_{j,q} \right)
$$
$$
= M_{1,2} \left( (K-1) M_{2,2}^2 + M_{2,4} \right) M_{x,2}^2 + M_{1^*,2} K M_{2,2} M_{2^*,2} M_{x,2}^2.
$$

This concludes the proof. $\square$

**Lemma 5.** *If $i \neq j$, $E(\langle \frac{\partial f_i}{\partial W_{2,1,q}}, \frac{\partial f_j}{\partial W_{2,1,q}} \rangle) = \Theta(d(K+d))$.*

*Proof.* Note that $\frac{\partial f_i}{\partial W_{2,1,q}} = (\hat{y}_i - y_i)W_{1,q,:}x_i = (W_2 W_1 - W_2^* W_1^*)x_i W_{1,q,:}x_i$. We have

$$
E(\langle \frac{\partial f_i}{\partial W_{2,1,q}}, \frac{\partial f_j}{\partial W_{2,1,q}} \rangle)
$$
$$
= E\left( (W_2 W_1 - W_2^* W_1^*)\, x_i W_{1,q,:}x_i \left( W_2 W_1 - W_2^* W_1^* \right) x_j W_{1,q,:}x_j \right)
$$
$$
= E\left( W_2 W_1 x_i W_{1,q,:}x_i W_2 W_1 x_j W_{1,q,:}x_j \right) + E\left( W_2^* W_1^* x_i W_{1,q,:}x_i W_2^* W_1^* x_j W_{1,q,:}x_j \right)
$$
$$
= E\left( \sum_{s=1}^{K} W_{2,1,s}^2 \left( W_{1,s,:}x_i W_{1,s,:}x_j \right) \left( W_{1,q,:}x_i W_{1,q,:}x_j \right) \right)
$$
$$
+ E\left( \sum_{s=1}^{K} {W_{2,1,s}^*}^2 \left( W_{1,s,:}^* x_i W_{1,s,:}^* x_j \right) \left( W_{1,q,:}x_i W_{1,q,:}x_j \right) \right)
$$

For the first term, we have

$$
E\left( \sum_{s=1}^{K} W_{2,1,s}^2 \left( W_{1,s,:}x_i W_{1,s,:}x_j \right) \left( W_{1,q,:}x_i W_{1,q,:}x_j \right) \right)
$$
$$
= E\left( \sum_{s=1}^{K} W_{2,1,s}^2 \left( \sum_{t=1}^{d} W_{1,s,t}x_{i,t} \right) \left( \sum_{t=1}^{d} W_{1,s,t}x_{j,t} \right) \left( \sum_{u=1}^{d} W_{1,q,u}x_{i,u} \right) \left( \sum_{u=1}^{d} W_{1,q,u}x_{j,u} \right) \right)
$$
$$
= \sum_{s=1}^{K} M_{2,2} E\left( \left( \sum_{t=1}^{d} W_{1,s,t}x_{i,t} \right) \left( \sum_{t=1}^{d} W_{1,s,t}x_{j,t} \right) \left( \sum_{u=1}^{d} W_{1,q,u}x_{i,u} \right) \left( \sum_{u=1}^{d} W_{1,q,u}x_{j,u} \right) \right).
$$

We now distinguish two cases. If $s \neq q$,

$$
E\left( \left( \sum_{t=1}^{d} W_{1,s,t}x_{i,t} \right) \left( \sum_{t=1}^{d} W_{1,s,t}x_{j,t} \right) \left( \sum_{u=1}^{d} W_{1,q,u}x_{i,u} \right) \left( \sum_{u=1}^{d} W_{1,q,u}x_{j,u} \right) \right)
$$
$$
= E\left( \left( \sum_{t=1}^{d} W_{1,s,t}^2 x_{i,t}x_{j,t} \right) \left( \sum_{u=1}^{d} W_{1,q,u}^2 x_{i,u}x_{j,u} \right) \right)
$$
$$
= E\left( \sum_{t=1}^{d} W_{1,s,t}^2 W_{1,q,t}^2 x_{i,t}^2 x_{j,t}^2 \right)
$$
$$
= d M_{1,2}^2 M_{x,2}^2.
$$

If $s = q$,

$$E\left(\left(\sum_{t=1}^{d} W_{1,s,t} x_{i,t}\right)\left(\sum_{t=1}^{d} W_{1,s,t} x_{j,t}\right)\left(\sum_{u=1}^{d} W_{1,q,u} x_{i,u}\right)\left(\sum_{u=1}^{d} W_{1,q,u} x_{j,u}\right)\right)$$

$$= E\left(\left(\sum_{t=1}^{d} W_{1,s,t}^2 x_{i,t} x_{j,t}\right)\left(\sum_{u=1}^{d} W_{1,q,u}^2 x_{i,u} x_{j,u}\right)\right)$$

$$+ E\left(\left(\sum_{t=1,t\neq v}^{d} W_{1,s,t} x_{i,t} W_{1,s,v} x_{j,v}\right)\left(\sum_{u=1,u\neq w}^{d} W_{1,q,u} x_{i,u} W_{1,q,w} x_{j,w}\right)\right)$$

$$= E\left(\sum_{t=1}^{d} W_{1,s,t}^4 x_{i,t}^2 x_{j,t}^2\right) + E\left(\left(\sum_{t=1,t\neq v}^{d} W_{1,s,t}^2 x_{i,t}^2 W_{1,s,v}^2 x_{j,v}^2\right)\right)$$

$$= d M_{1,4} M_{x,2}^2 + d(d-1) M_{1,2}^2 M_{x,2}^2.$$

Combining the two cases,

$$E\left(\sum_{s=1}^{K} W_{2,1,s}^2 \left(W_{1,s,:} x_i\right)^2 \left(W_{1,q,:} x_i\right)^2\right) = M_{2,2}\left((K-1)\, d M_{1,2}^2 M_{x,2}^2 + d M_{1,4} M_{x,2}^2 + d(d-1) M_{1,2}^2 M_{x,2}^2\right)$$

$$= \Theta(Kd + d^2)$$

For the second term, we have

$$E\left(\sum_{s=1}^{K} {W_{2,1,s}^*}^2 \left(W_{1,s,:}^* x_i W_{1,s,:}^* x_j\right)\left(W_{1,q,:} x_i W_{1,q,:} x_j\right)\right)$$

$$= E\left(\sum_{s=1}^{K} {W_{2^*,1,s}}^2 \left(\sum_{t=1}^{d} W_{1^*,s,t} x_{i,t}\right)\left(\sum_{t=1}^{d} W_{1^*,s,t} x_{j,t}\right)\left(\sum_{u=1}^{d} W_{1,q,u} x_{i,u}\right)\left(\sum_{u=1}^{d} W_{1,q,u} x_{j,u}\right)\right)$$

$$= E\left(\sum_{s=1}^{K} {W_{2^*,1,s}}^2 \left(\sum_{t=1}^{d} W_{1^*,s,t}^2 x_{i,t} x_{j,t}\right)\left(\sum_{u=1}^{d} W_{1,q,u}^2 x_{i,u} x_{j,u}\right)\right)$$

$$= E\left(\sum_{s=1}^{K} {W_{2^*,1,s}}^2 \left(\sum_{t=1}^{d} W_{1^*,s,t}^2 W_{1,q,u}^2 x_{i,t}^2 x_{j,t}^2\right)\right)$$

$$= K d M_{2^*,2} M_{1^*,2} M_{1,2} M_{x,2}^2 = \Theta(Kd).$$

Combing the first term and the second term, we obtain the desired result. $\qquad\square$

By applying Proposition A.1 and linearity of expectation, we obtain the following result:

**Theorem 4.** *We have*

$$E\left(\sum_{i=1}^{n} ||\nabla f_i||^2\right) = \Theta(nK^2 d^2) \tag{A.1}$$

$$E\left(\sum_{i=1, j\neq i}^{n} \langle \nabla f_i, \nabla f_j\rangle\right) = \Theta(n^2 K^2 d + n^2 K d^2) \tag{A.2}$$

The above theorem computes the expectation of each term of the ratio. In order to obtain a result on the expectation of the ratio, we also need to show that the value of each term will be concentrated around the expectation with high probability. To prove such a result, we first compute the variance.

**Theorem 5.** *We have*

$$Var\left(\sum_{i=1}^{n} ||\nabla f_i||^2\right) = \Theta(n^2 K^4 d^3) \tag{A.3}$$

$$Var\left(\sum_{i=1, j\neq i}^{n} \langle \nabla f_i, \nabla f_j\rangle\right) = \Theta(n^4 K^3 d^2) \tag{A.4}$$

**Theorem 1.** *Consider a 2 LNN. Let the weights $W_{l,p,q}, W_{l,p,q}^*$ for $l \in \{1,2\}$ and $\mathbf{x}_i$ be independently drawn random variables, such that their k-th order moments for $k \leq 4$ are bounded in a postive interval. Then, with arbitrary constant probability, the following holds:*

$$B_{\mathcal{S}}(\mathbf{w}) \geq \frac{\Theta(nKd)}{\Theta(Kn + dn + Kd)}$$

*Proof.* By Chebyshev's Inequality, we have

$$\Pr\left(\,|\sum_{i=1}^{n}||\nabla f_i||^2 - E\left(\sum_{i=1}^{n}||\nabla f_i||^2\right)|\geq \epsilon\right) \leq \frac{\text{Var}(\sum_{i=1}^{n}||\nabla f_i||^2)}{\epsilon^2}$$

Using the above two theorems, and choosing parameter $\epsilon = \Theta(nK^2 d^{3/2}\delta^{-1/2})$, we have that with probability $1 - \delta$,

$$\Theta(nK^2 d^2) - \Theta(nK^2 d^{3/2}\delta^{-1/2}) \leq \sum_{i=1}^{n}||\nabla f_i||^2 \leq \Theta(nK^2 d^2) + \Theta(nK^2 d^{3/2}\delta^{-1/2})$$

We can similarly use Chebyshev's inequality to obtain that with probability $1 - \delta$,

$$\sum_{i=1,j\neq i}^{n}\langle \nabla f_i, \nabla f_j\rangle \leq \Theta(n^2 Kd(K+d)) + \Theta(n^2 K^{3/2} d\delta^{-1/2})$$

By applying the union bound, we can now bound the ratio as desired:

$$\frac{n\sum_{i=1}^{n}||\nabla f_i||^2}{||\sum_{i=1}^{n}\nabla f_i||^2} = \frac{n\sum_{i=1}^{n}||\nabla f_i||^2}{\sum_{i=1,j\neq i}^{n}\langle \nabla f_i, \nabla f_j\rangle + \sum_{i=1}^{n}||\nabla f_i||^2}$$

$$\geq \frac{\Theta(n^2 K^2 d^2) - \Theta(\frac{n^2 K^2 d^{3/2}}{\sqrt{\delta}})}{\Theta(n^2 Kd(K+d)) + \Theta(\frac{n^2 K^{3/2} d}{\sqrt{\delta}}) + \Theta(nK^2 d^2) - \Theta(\frac{nK^2 d^{3/2}}{\sqrt{\delta}})}$$

$$= \frac{\Theta(nKd)}{\Theta(Kn + dn + Kd)}$$

Here we assumed that $\delta$ is chosen to be some arbitrarily small constant. $d$ and $n$ is sufficiently large such that $d > \frac{1}{\delta}$ and $\Theta(n^2)$ dominates $\Theta(n)$. $\qquad\square$

# B    Proofs for 2-Layer Nonlinear Neural Networks

In this section we present the detailed proof of Theorem 2, which is restated as below (A few constants are stated explicitly for ease of proof.).

**Theorem 2.** *Consider a 2-layer NN with a monotone activation function $\sigma$ such that for every $x$ we have: $-\sigma(x) = \sigma(-x)$, and both $|\sigma(x)|$ and $\sup_x\{x\sigma'(x)\}$ are bounded. Let the weights $W_{l,p,q}, W^*_{l,p,q}$ for $l \in \{1, 2\}$ and $\mathbf{x}_i$ be i.i.d. random variables from $\mathcal{N}(0, 1)$. Then, with high probability, the following holds:*

$$\frac{\mathbb{E}[n\sum_{i=1}^{n}||\nabla f_i||_2^2]}{\mathbb{E}[||\sum_{i=1}^{n}\nabla f_i||_2^2]} \geq \Omega(\frac{Kd^2}{Kd + K + d}).$$

*where the expectation is over $W_2, W_2^*$.*

Denote the bound of $|\sigma(x)|$ by $c_{max}$, and the bound of $\sup_x\{x\sigma'(x)\}$ by $c_{sup}$.

## B.1    Notations and Models

We consider a 2-later nonlinear neural network. Let $W_1, W_2$ be the coefficient matrix of the first and second layer, respectively. $W_{a,p,q}$ is the $p, q$ element in matrix $a$. For ease of notations, let us further define

$$A_1 = n\sum_{i=1}^{n}(\hat{y}_i - y_i)^2 \left(\sum_{p=1}^{K}\sum_{q=1}^{d}\left(\frac{\partial \hat{y}_i}{\partial W_{1,p,q}}\right)^2\right),$$

$$A_2 = n\sum_{i=1}^{n}(\hat{y}_i - y_i)^2 \left(\sum_{q=1}^{K}\left(\frac{\partial \hat{y}_i}{\partial W_{2,1,q}}\right)^2\right),$$

$$B_1 = \left(\sum_{p=1}^{K}\sum_{q=1}^{d}\left(\sum_{i=1}^{n}(\hat{y}_i - y_i)\frac{\partial \hat{y}_i}{\partial W_{1,p,q}}\right)^2\right).$$

$$B_2 = \left(\sum_{q=1}^{K}\left(\sum_{i=1}^{n}(\hat{y}_i - y_i)\frac{\partial \hat{y}_i}{\partial W_{2,1,q}}\right)^2\right).$$

## B.2 Some Helper Lemmas

We first provide some lemmas.

**Lemma 6.** *Let $X, Y, Z$ be three normal distribution. Let $\rho_{XY}, \rho_{YZ}, \rho_{XZ}$ be the correlation between those random variables. Let $f()$ be a monotone, bounded, and differentiable function. More precisely, $f(x) \geq f(y)$ iff $x \geq y$, $|f(x)| \leq f_{\max}$. Further assume $\sup f'(x)x = c$. Then we have*

$$|E(f(X)f'(Y)Z)| \leq (1 - \rho_{XY}^2)^{-1}|\rho_{XZ} - \rho_{YZ}\rho_{XY}| \times f_{max}f'_{max}\sigma_x + (1 - \rho_{XY}^2)^{-1}|-\rho_{XZ}\rho_{XY} + \rho_{YZ}| \times f_{max}c.$$

*Proof.* Given $X$ and $Y$, random variable $Z$ is a normal distributed random variable with mean $E(Z|X,Y) = (1 - \rho_{XY}^2)^{-1}(\rho_{XZ} - \rho_{YZ}\rho_{XY})X + (1 - \rho_{XY}^2)^{-1}(-\rho_{XZ}\rho_{XY} + \rho_{YZ})Y$. Thus it holds that

$$
\begin{aligned}
&|E(f(X)f'(Y)Z)|\\
=&|E(E(Z|X,Y)f(X)f'(Y)|\\
=&(1 - \rho_{XY}^2)^{-1}|E((\rho_{XZ} - \rho_{YZ}\rho_{XY})Xf(Xf'(Y)) + (-\rho_{XZ}\rho_{XY} + \rho_{YZ})Yf(X)f'(Y))|\\
\leq&(1 - \rho_{XY}^2)^{-1}|\rho_{XZ} - \rho_{YZ}\rho_{XY}| \times E|Xf(X)f'(Y)| + (1 - \rho_{XY}^2)^{-1}|-\rho_{XZ}\rho_{XY} + \rho_{YZ}| \times E|Yf(X)f'(Y)|\\
\leq&(1 - \rho_{XY}^2)^{-1}|\rho_{XZ} - \rho_{YZ}\rho_{XY}| \times f_{max}f'_{max}\sigma_x + (1 - \rho_{XY}^2)^{-1}|-\rho_{XZ}\rho_{XY} + \rho_{YZ}| \times f_{max}c.
\end{aligned}
$$

$\square$

**Lemma 7.** *Let $a_1, a_2, \cdots, a_d, b_1, b_2, \cdots, b_d$ be i.i.d. standard normal distribution. Then we have w.p. $1 - 3\delta$,*

$$\frac{\sum a_i b_i}{\sqrt{\sum_i a_i^2}\sqrt{\sum_i b_i^2}} \leq \frac{\sqrt{d}\log\frac{2}{\delta}}{d - \sqrt{d}\log\frac{2}{\delta}} \tag{B.1}$$

*Proof.* Directly applying Chernoff bound for normal distribution. $\square$

**Lemma 8.** *Let $Z_1, Z_2$ be two r.v.s with normal distribution. Let $\rho = \frac{V_{12}}{\sigma_1\sigma_2}$, where $V_{12}$ is the correlation between $Z_1, Z_2$. Consider a function $\sigma(\cdot)$ such that $\sigma(x) = -\sigma(-x)$, $|\sigma(\cdot)| \leq \sigma_{max}$, and $\sup_x \sigma(x)x = \alpha_G$. Then we have*

$$|E(\sigma(Z_1)\sigma(Z_2))| \leq \left(\frac{\sigma_{\max} + 2\sqrt{\rho}\sigma_{\max} + 4\alpha_G\sqrt{1-\rho}}{\sqrt{1-\rho^2}}\right)\sigma_{\max}\sqrt{\rho}.$$

*Proof.* Expanding the expectation, we have

$$
\begin{aligned}
E(\sigma(Z_1)\sigma(Z_2)) &= \int_{-\infty}^{+\infty} \sigma(z_1)\sigma(z_2)\frac{1}{2\pi\sigma_1\sigma_2\sqrt{1-\rho^2}}\exp\left(-\frac{\frac{1}{\sigma_1^2}z_1^2 - \frac{2\rho}{\sigma_1\sigma_2}z_1z_2 + \frac{1}{\sigma_2^2}z_2^2}{2(1-\rho^2)}\right)dz_1dz_2\\
&= \int_{-\infty}^{+\infty} \sigma(z_1)\sigma(z_2)\frac{1}{2\pi\sigma_1\sigma_2\sqrt{1-\rho^2}}\exp\left(-\frac{\frac{1}{\sigma_1^2}(z_1 - \frac{\rho\sigma_1}{\sigma_2}z_2)^2 + \frac{1-\rho^2}{\sigma_2^2}z_2^2}{2(1-\rho^2)}\right)dz_1dz_2\\
&= \int_{-\infty}^{+\infty} \sigma(z_3 + \frac{\rho\sigma_1}{\sigma_2}z_2)\sigma(z_2)\frac{1}{2\pi\sigma_1\sigma_2\sqrt{1-\rho^2}}\exp\left(-\frac{\frac{1}{\sigma_1^2}(z_3)^2 + \frac{1-\rho^2}{\sigma_2^2}z_2^2}{2(1-\rho^2)}\right)dz_3dz_2\\
&= \int_{-\infty}^{+\infty} \sigma(\sigma_1 u_3 + \rho\sigma_1 u_2)\sigma(\sigma_2 u_2)\frac{1}{2\pi\sqrt{1-\rho^2}}\exp\left(-\frac{u_3^2 + (1-\rho^2)u_2^2}{2(1-\rho^2)}\right)du_3du_2
\end{aligned}
$$

where we simply change the integration variable. Let $G(u) = \sigma(\sigma_1 u)$. Note that $|G(u)| \leq \sigma_{max}$. The above integration becomes

$$\int_{-\infty}^{+\infty} G(u_3 + \rho u_2)G(\frac{\sigma_2}{\sigma_1}u_2)\frac{1}{2\pi\sqrt{1-\rho^2}}\exp\left(-\frac{u_3^2 + (1-\rho^2)u_2^2}{2(1-\rho^2)}\right)du_3du_2.$$

Now First fix $u_2$ and decompose the integration into two parts. The first part is the integration on $(-\infty, -x) \cap (x, +\infty)$ and the second part is the integration on $[-x, x]$. First consider the case $u_2 \geq 0$. For the first part,

$$\int_{-\infty}^{-x} + \int_{x}^{+\infty} G(u_3 + \rho u_2) G(\frac{\sigma_2}{\sigma_1} u_2) \frac{1}{2\pi\sqrt{1-\rho^2}} \exp\left(-\frac{u_3^2 + (1-\rho^2) u_2^2}{2(1-\rho^2)}\right) du_3$$

$$= \int_{x}^{+\infty} \left(G(u_3 + \rho u_2) + G(-u_3 + \rho u_2)\right) G(\frac{\sigma_2}{\sigma_1} u_2) \frac{1}{2\pi\sqrt{1-\rho^2}} \exp\left(-\frac{u_3^2 + (1-\rho^2) u_2^2}{2(1-\rho^2)}\right) du_3$$

$$= \int_{x}^{+\infty} \left(G(u_3 + \rho u_2) - G(u_3 - \rho u_2)\right) G(\frac{\sigma_2}{\sigma_1} u_2) \frac{1}{2\pi\sqrt{1-\rho^2}} \exp\left(-\frac{u_3^2 + (1-\rho^2) u_2^2}{2(1-\rho^2)}\right) du_3,$$

where the last inequality is by the symmetry of the function $G$, i.e., $G(x) = -(G-x)$. Note that

$$|G'(y)| = |\sigma(\sigma_1 y)\sigma_1 y\frac{1}{y}| \leq \alpha_G \frac{1}{y}.$$

Let $x = \rho u_2 + w$, where $w \geq 0$. Then we have for all $u_3 \geq x$,

$$|G'(u_3 + \rho u_2)| \leq \alpha_G \frac{1}{u_3 + \rho u_2} \leq \alpha_G \frac{1}{w}$$

and

$$|G'(u_3 - \rho u_2)| \leq \alpha_G \frac{1}{u_3 - \rho u_2} \leq \alpha_G \frac{1}{w},$$

which in fact proves $G(y)$ is Lipschitz continuous with constant $\frac{\alpha_G}{w}$ for $y \geq w$. Thus, we now have

$$|G(u_3 + \rho u_2) - G(u_3 - \rho u_2)| \leq \alpha_G \frac{1}{w} 2\rho u_2.$$

Since $G$ is monotone, we have

$$G(u_3 + \rho u_2) - G(u_3 - \rho u_2) \leq \alpha_G \frac{1}{w} 2\rho u_2.$$

Apply this inequality in the integration, we have

$$\int_{x}^{+\infty} \left(G(u_3 + \rho u_2) - G(u_3 - \rho u_2)\right) G(\frac{\sigma_2}{\sigma_1} u_2) \frac{1}{2\pi\sqrt{1-\rho^2}} \exp\left(-\frac{u_3^2 + (1-\rho^2) u_2^2}{2(1-\rho^2)}\right) du_3$$

$$\leq \int_{x}^{+\infty} \left(\frac{\alpha_G}{w} 2\rho u_2\right) G(\frac{\sigma_2}{\sigma_1} u_2) \frac{1}{2\pi\sqrt{1-\rho^2}} \exp\left(-\frac{u_3^2 + (1-\rho^2) u_2^2}{2(1-\rho^2)}\right) du_3$$

$$\leq \int_{-\infty}^{+\infty} \left(\frac{\alpha_G}{w} 2\rho u_2\right) G(\frac{\sigma_2}{\sigma_1} u_2) \frac{1}{2\pi\sqrt{1-\rho^2}} \exp\left(-\frac{u_3^2 + (1-\rho^2) u_2^2}{2(1-\rho^2)}\right) du_3$$

$$= \left(\frac{\alpha_G}{w} 2\rho u_2\right) G(\frac{\sigma_2}{\sigma_1} u_2) \frac{1}{\sqrt{2\pi}} \exp\left(-\frac{u_2^2}{2}\right).$$

Now let us consider the second part of the integration.

$$\int_{-x}^{x} G(u_3 + \rho u_2) G(\frac{\sigma_2}{\sigma_1} u_2) \frac{1}{2\pi\sqrt{1-\rho^2}} \exp\left(-\frac{u_3^2 + (1-\rho^2) u_2^2}{2(1-\rho^2)}\right) du_3$$

$$\leq \int_{-x}^{x} \sigma_{\max} G(\frac{\sigma_2}{\sigma_1} u_2) \frac{1}{2\pi\sqrt{1-\rho^2}} \exp\left(-\frac{u_3^2 + (1-\rho^2) u_2^2}{2(1-\rho^2)}\right) du_3$$

$$\leq \int_{-x}^{x} \sigma_{\max} G(\frac{\sigma_2}{\sigma_1} u_2) \frac{1}{\sqrt{2\pi}\sqrt{1-\rho^2}} \exp\left(-\frac{u_2^2}{2}\right) du_3$$

$$= 2x\sigma_{max} G(\frac{\sigma_2}{\sigma_1} u_2) \frac{1}{\sqrt{2\pi}\sqrt{1-\rho^2}} \exp\left(-\frac{u_2^2}{2}\right),$$

where the first inequality is because $\sigma_{\max} \geq G$, the second inequality is because $exp(-a^2) \leq 1$. Combing the the integration, we finally have

$$\int_{-\infty}^{+\infty} G(u_3 + \rho u_2) G(\frac{\sigma_2}{\sigma_1} u_2) \frac{1}{2\pi\sqrt{1-\rho^2}} \exp\left(-\frac{u_3^2 + (1-\rho^2) u_2^2}{2(1-\rho^2)}\right) du_3$$

$$\leq \left(\frac{\alpha_G}{w} 2\rho u_2\right) G(\frac{\sigma_2}{\sigma_1} u_2) \frac{1}{\sqrt{2\pi}} \exp\left(-\frac{u_2^2}{2}\right) + 2x\sigma_{max} G(\frac{\sigma_2}{\sigma_1} u_2) \frac{1}{\sqrt{2\pi}\sqrt{1-\rho^2}} \exp\left(-\frac{u_2^2}{2}\right),$$

where $x = \rho u_2 + w$. Let $w = \frac{\alpha_G 2u_2 \sqrt{\rho}\sqrt{1-\rho}}{\sigma_{max}}$. We have

$$\int_{-\infty}^{+\infty} G(u_3 + \rho u_2)G(\frac{\sigma_2}{\sigma_1}u_2)\frac{1}{2\pi\sqrt{1-\rho^2}}\exp\left(-\frac{u_3^2 + \left(1-\rho^2\right)u_2^2}{2(1-\rho^2)}\right)du_3$$

$$\left(\frac{\alpha_G}{w}2\rho u_2\right)G(\frac{\sigma_2}{\sigma_1}u_2)\frac{1}{\sqrt{2\pi}}\exp\left(-\frac{u_2^2}{2}\right) + 2x\sigma_{max}G(\frac{\sigma_2}{\sigma_1}u_2)\frac{1}{\sqrt{2\pi}\sqrt{1-\rho^2}}\exp\left(-\frac{u_2^2}{2}\right)$$

$$= \left(\frac{\sqrt{\rho}\sigma_{max}}{\sqrt{1-\rho^2}}\right)G(\frac{\sigma_2}{\sigma_1}u_2)\frac{1}{\sqrt{2\pi}}\exp\left(-\frac{u_2^2}{2}\right)$$

$$+ 2(\rho u_2 + \frac{\alpha_G 2u_2\sqrt{\rho}\sqrt{1-\rho}}{\sigma_{max}})\sigma_{max}G(\frac{\sigma_2}{\sigma_1}u_2)\frac{1}{\sqrt{2\pi}\sqrt{1-\rho^2}}\exp\left(-\frac{u_2^2}{2}\right)$$

$$\leq \left(\frac{\sqrt{\rho}\sigma_{max}^2}{\sqrt{1-\rho^2}}\right)\frac{1}{\sqrt{2\pi}}\exp\left(-\frac{u_2^2}{2}\right) + 2(\rho u_2 + \frac{\alpha_G 2u_2\sqrt{\rho}\sqrt{1-\rho}}{\sigma_{max}})\sigma_{max}^2\frac{1}{\sqrt{2\pi}\sqrt{1-\rho^2}}\exp\left(-\frac{u_2^2}{2}\right)$$

$$= \left(\frac{\sqrt{\rho}\sigma_{max}^2 + 2\rho u_2\sigma_{max}^2 + 4\alpha_G u_2\sqrt{\rho}\sigma_{max}\sqrt{1-\rho}}{\sqrt{1-\rho^2}}\right)\frac{1}{\sqrt{2\pi}}\exp\left(-\frac{u_2^2}{2}\right),$$

where the first equation is because we plug in the expression of $w$, the inequality is due to $G \leq \sigma_{max}$. Similarly we can prove it for the case when $u \leq 0$,

$$\int_{-\infty}^{+\infty} G(u_3 + \rho u_2)G(\frac{\sigma_2}{\sigma_1}u_2)\frac{1}{2\pi\sqrt{1-\rho^2}}\exp\left(-\frac{u_3^2 + \left(1-\rho^2\right)u_2^2}{2(1-\rho^2)}\right)du_3$$

$$\leq \left(\frac{\sqrt{\rho}\sigma_{max}^2 + 2\rho(-u_2)\sigma_{max}^2 + 4\alpha_G(-u_2)\sqrt{\rho}\sigma_{max}\sqrt{1-\rho}}{\sqrt{1-\rho^2}}\right)\frac{1}{\sqrt{2\pi}}\exp\left(-\frac{u_2^2}{2}\right).$$

Thus, we have

$$E(\sigma(Z_1)\sigma(Z_2)) \leq \int_0^{+\infty}\left(\frac{\sqrt{\rho}\sigma_{max}^2 + 2\rho u_2\sigma_{max}^2 + 4\alpha_G u_2\sqrt{\rho}\sigma_{max}\sqrt{1-\rho}}{\sqrt{1-\rho^2}}\right)\frac{1}{\sqrt{2\pi}}\exp\left(-\frac{u_2^2}{2}\right)du_2$$

$$= \left(\frac{\sqrt{\rho}\sigma_{max}^2 + 2\rho\sigma_{max}^2 + 4\alpha_G\sqrt{\rho}\sigma_{max}\sqrt{1-\rho}}{\sqrt{1-\rho^2}}\right)$$

$$= \left(\frac{\sigma_{max} + 2\sqrt{\rho}\sigma_{max} + 4\alpha_G\sqrt{1-\rho}}{\sqrt{1-\rho^2}}\right)\sigma_{max}\sqrt{\rho}.$$

By symmetry, we have

$$-E(\sigma(Z_1)\sigma(Z_2)) \leq \left(\frac{\sigma_{max} + 2\sqrt{\rho}\sigma_{max} + 4\alpha_G\sqrt{1-\rho}}{\sqrt{1-\rho^2}}\right)\sigma_{max}\sqrt{\rho},$$

which completes the proof. $\square$

## B.3  Main Proof

The main proof of the theorem consists of 4 lemmas, based on which the main theorem becomes straightforward.

**Lemma 9.** *Suppose $W, W^*, c_i$ are all i.i.d. random variables sampled from standard normal distribution. Then w.h.p,*

$$E_{W_2,W_2^*}A_1 \geq \mathcal{O}(n^2 K^2 d).$$

*Proof.* Expanding the expression of $A_1$, we have

$$E_{W_2}A_1 = E_{W_2}\left(n\sum_{i=1}^{n}(\hat{y}_i - y_i)^2\left(\sum_{p=1}^{K}\sum_{q=1}^{d}\left(\frac{\partial \hat{y}_i}{\partial W_{1,p,q}}\right)^2\right)\right)$$

$$= E_{W_2}\left(n\sum_{i=1}^{n}(W_2\sigma(W_1 x_i) - W_2^*\sigma(W_1^* x_i))^2\left(\sum_{p=1}^{K}\sum_{q=1}^{d}\left(W_{2,1,p}\sigma'(W_{1,p,:}x_i)x_{i,q}\right)^2\right)\right)$$

$$= nE_{W_2}\left(\sum_{i=1}^{n}(W_2\sigma(W_1 x_i) - W_2^*\sigma(W_1^* x_i))^2\left(\sum_{p=1}^{K}\sum_{q=1}^{d}\left(W_{2,1,p}\sigma'(W_{1,p,:}x_i)x_{i,q}\right)^2\right)\right)$$

$$= nE_{W_2}\left(\sum_{i=1}^{n}\sum_{r=1}^{K}(W_{2,1,r}^2\sigma(W_1 x_i)^2 + W_{2,1,r,*}^2\sigma(W_1^* x_i)^2)\left(\sum_{p=1}^{K}\sum_{q=1}^{d}\left(W_{2,1,p}\sigma'(W_{1,p,:}x_i)x_{i,q}\right)^2\right)\right)$$

$$\geq n\left(\sum_{i=1}^{n}\sum_{r=1}^{K}(\sigma(W_{1,r,:}x_i)^2 + \sigma(W_{1,r,:}^* x_i)^2)\left(\sum_{p=1}^{K}\sum_{q=1}^{d}\left(\sigma'(W_{1,p,:}x_i)x_{i,q}\right)^2\right)\right).$$

Since $\sigma(\cdot)$ is bounded by $\sigma_{\max}$, we have

$$Pr(|\sigma(W_{1,r,:}x_i)^2\sigma'(W_{1,p:}x_i)^2 x_{i,q}^2| \geq t) \leq Pr(|\sigma_{max}^2\sigma_{max}'^2 x_{i,q}^2| \geq t)$$

$$\leq 2\exp(-\frac{t^2}{2\sigma_{max}^4\sigma_{max}'^4})$$

where the last equation is due to the fact that $x_{i,q}$ is standard normal distributed. This implies $\sigma(W_{1,r,:}x_i)^2\sigma'^2(W_{1,p:}x_i)x_{i,q}^2$ is sub-exponential (where $x_{i,q}^2$ is chi-square). Thus, we can apply Bernstein inequality to $\sigma(W_{1,r,:}x_i)^2\sigma'^2(W_{1,p:}x_i)x_{i,q}^2$, to obtain w.p. $1 - \delta$,

$$\sum_{i=1}^{n}\sigma(W_{1,r,:}x_i)^2\sigma'^2(W_{1,p:}x_i)x_{i,q}^2 - E_{x_i}\sum_{i=1}^{n}\sigma(W_{1,r,:}x_i)^2\sigma'^2(W_{1,p:}x_i)x_{i,q}^2 \geq -\sqrt{n}\sigma_{max}^2\sigma_{max}'^2\log\frac{2}{\delta}$$

and similarly w.p. $1 - \delta$,

$$|\sum_{i=1}^{n}\sigma(W_{1,r,:}^* x_i)^2\sigma'^2(W_{1,p:}x_i)x_{i,q}^2 - E_{x_i}\sum_{i=1}^{n}\sigma(W_{1,r,:}^* x_i)^2\sigma'^2(W_{1,p:}x_i)x_{i,q}^2 \geq -\sqrt{n}\sigma_{max}^2\sigma_{max}'^2\log\frac{2}{\delta}$$

By union bound, we have w.p $1 - 2\delta$, the above two are both true. Plug them in the expression of $A_1$. Finally, we have w.p. $1 - 2\delta$,

$$E_{W_2}A_1 \geq n\left(\sum_{i=1}^{n}\sum_{r=1}^{K}(\sigma(W_{1,r,:}x_i)^2 + \sigma(W_{1,r,:}^* x_i)^2)\left(\sum_{p=1}^{K}\sum_{q=1}^{d}\left(\sigma'(W_{1,p,:}x_i)x_{i,q}\right)^2\right)\right)$$

$$= n\sum_{p=1}^{K}\sum_{q=1}^{d}\sum_{r=1}^{K}\sum_{i=1}^{n}\left(\sigma(W_{1,r,:}x_i)^2 + \sigma(W_{1,r,:}^* x_i)^2\right)\left(\sigma'(W_{1,p,:}x_i)x_{i,q}\right)^2$$

$$= n\sum_{p=1}^{K}\sum_{q=1}^{d}\sum_{r=1}^{K}\sum_{i=1}^{n}\left(\sigma(W_{1,r,:}x_i)^2\left(\sigma'(W_{1,p,:}x_i)x_{i,q}\right)^2 + \sigma(W_{1,r,:}^* x_i)^2\left(\sigma'(W_{1,p,:}x_i)x_{i,q}\right)^2\right)$$

$$\geq 2n^2\left(\sum_{p=1}^{K}\sum_{q=1}^{d}\sum_{r=1}^{K}E_{x_i}\sigma(W_{1,r,:}x_i)^2\sigma'^2(W_{1,p:}x_i)x_{i,q}^2 + E_{x_i}\sigma(W_{1,r,:}^* x_i)^2\sigma'^2(W_{1,p:}x_i)x_{i,q}^2 - a_0\right)$$

$$\geq 2n^2 d\left(\sum_{p=1}^{K}\sum_{r=1}^{K}E_{x_i}\sigma(W_{1,r,:}x_i)^2\sigma'^2(W_{1,p:}x_i)x_{i,q}^2 + E_{x_i}\sigma(W_{1,r,:}^* x_i)^2\sigma'^2(W_{1,p:}x_i)x_{i,q}^2 - a_0\right)$$

where $a_0 = 2\frac{1}{\sqrt{n}}\sigma_{max}^2\sigma_{max}'^2\log\frac{2}{\delta}$ is the extra error term. Note that this term is small and typically can be ignored.

Since the term within summation is bounded, we can apply Hoeffding bound over $p$ and $r$ separately. Finally we will get w.p. $1 - 6\delta$,

$$E_{W_2}A_1 \geq 2n^2 K^2 d\left(E\sigma(W_{1,r,:}x_i)^2\sigma'^2(W_{1,p:}x_i)x_{i,q}^2 + E\sigma(W_{1,r,:}^* x_i)^2\sigma'^2(W_{1,p:}x_i)x_{i,q}^2 - O(\frac{1}{\sqrt{n}} + \frac{1}{\sqrt{K}})\log\frac{1}{\delta}\right)$$

$$= O(n^2 K^2 d)$$

which completes the proof. $\square$

**Lemma 10.** *Suppose $W, W^*, x_i$ are all i.i.d. random variables sampled from standard normal distribution. Then w.h.p,*

$$E_{W_2, W_2^*} A_2 \geq \mathcal{O}(n^2 K^2).$$

*Proof.*

$$E_{W_2, W_2^*} A_2 = E_{W_2} \left( n \sum_{i=1}^{n} (\hat{y}_i - y_i)^2 \left( \sum_{q=1}^{K} \left( \frac{\partial \hat{y}_i}{\partial W_{2,1,q}} \right)^2 \right) \right)$$

$$= E_{W_2} \left( n \sum_{i=1}^{n} ||W_2 \sigma(W_1 x_i) - W_2^* \sigma(W_1^* x_i)||^2 \left( \sum_{q=1}^{K} (\sigma(W_{1,q,:} x_i))^2 \right) \right)$$

$$= n \sum_{i=1}^{n} \sum_{r=1}^{K} (\sigma(W_{1,r,:} x_i)^2 + \sigma(W_{1,r,:}^* x_i)^2) \left( \sum_{q=1}^{K} (\sigma(W_{1,q,:} x_i))^2 \right).$$

Now fix $W_1$ and thus $r$. Note that $|| \left( \sigma(W_{1,r,:} x_i)^2 + \sigma(W_{1,r,:}^* x_i)^2 \right) \sum_{q=1}^{K} (\sigma(W_{1,q,:} x_i))^2 || \leq 2K\sigma_{\max}^4$ Applying Hoeffding bound to the term in the summation over the randomness of $x_i$, we have w.p. $(1 - \delta)$,

$$\left| \sum_{i=1}^{n} (\sigma(W_{1,r,:} x_i)^2 + \sigma(W_{1,r,:}^* x_i)^2) \left( \sum_{q=1}^{K} (\sigma(W_{1,q,:} x_i))^2 \right) - nE_{x_i} (\sigma(W_{1,r,:} x_i)^2 \right.$$

$$\left. + \sigma(W_{1,r,:}^* x_i)^2) \left( \sum_{q=1}^{K} (\sigma(W_{1,q,:} x_i)|)^2 \right) \right| \leq \log \frac{2}{\delta} \sqrt{8n} K \sigma_{max}^4.$$

And thus we have w.p. $1 - \delta$,

$$E_{W_2, W_2^*} A_2 \geq n \sum_{r=1}^{K} \left( nE_{x_i} (\sigma(W_{1,r,:} x_i)^2 + \sigma(W_{1,r,:}^* x_i)^2) \left( \sum_{q=1}^{K} \left( \sigma(W_{1,q,:} x_i)^2 \right) \right) - \log \frac{2}{\delta} \sqrt{8n} K \sigma_{max}^4 \right)$$

$$= n^2 \sum_{r=1}^{K} \left( E_{x_i} (\sigma(W_{1,r,:} x_i)^2 + \sigma(W_{1,r,:}^* x_i)^2) \left( \sum_{q=1}^{K} \left( \sigma(W_{1,q,:} x_i)^2 \right) \right) - \log \frac{2}{\delta} \sqrt{\frac{8}{n}} K \sigma_{max}^4 \right)$$

$$\geq n^2 \sum_{r=1}^{K} \left( E_{x_i} (\sigma(W_{1,r,:} x_i)^2) \left( \sum_{q=1}^{K} \left( \sigma(W_{1,q,:} x_i)^2 \right) \right) - \log \frac{2}{\delta} \sqrt{\frac{8}{n}} K \sigma_{max}^4 \right)$$

$$\geq n^2 \left( \left( \sum_{r=1}^{K} E_{x_i} (\sigma(W_{1,r,:} x_i)^2) \right) \left( \sum_{q=1}^{K} \left( \sigma(W_{1,q,:} x_i)^2 \right) \right) - \log \frac{2}{\delta} \sqrt{\frac{8}{n}} K^2 \sigma_{max}^4 \right).$$

Now let us apply Hoffding bound over $W_1$, we have w.p. $1 - \delta$,

$$\left| \sum_{r=1}^{K} \sigma^2(W_{1,r,:} x_i) - KE_{W_1} \sigma^2(W_{1,1,:} x_i) \right| \leq \log \frac{2}{\delta} \sqrt{K} \sigma_{max}^2,$$

and similarly w.p. $1 - \delta$,

$$\left| \sum_{r=1}^{K} \sigma^2(W_{1,q,:} x_i) - KE_{W_1} \sigma^2(W_{1,1,:} x_i) \right| \leq \log \frac{2}{\delta} \sqrt{K} \sigma_{max}^2.$$

Thus, w.p. $1 - 3\delta$, we have

$$E_{W_2, W_2^*} A_2 \geq n^2 K^2 E_x \left( \left( E_{W_1} \sigma^2(W_{1,1,:} x_i) - \log \frac{2}{\delta} \sigma_{max}^2 \sqrt{\frac{1}{K}} \right)^2 - \log \frac{2}{\delta} \sqrt{8n} K^2 \sigma_{max}^4 \right)$$

$$= n^2 K^2 E_x (E_{W_1} \sigma^2(W_{1,1,:} x_i))^2 - E_{W_1,x} \sigma^2(W_{1,1,:} x_i)(\log \frac{2}{\delta} \sigma_{max}^2 \sqrt{K}) K n^2$$

$$+ n^2 (\log \frac{2}{\delta})^2 K \sigma_{max}^4 - \log \frac{2}{\delta} \sqrt{8n} K \sigma_{max}^4$$

$$\geq n^2 K^2 E_x (E_{W_1} \sigma^2(W_{1,1,:} x_i))^2 - (\log \frac{2}{\delta} \sigma_{max}^4 \sqrt{K}) K n^2$$

$$+ n^2 (\log \frac{2}{\delta})^2 K \sigma_{max}^4 - \log \frac{2}{\delta} \sqrt{8n} K \sigma_{max}^4.$$

Changing $\delta$ to $\frac{\delta}{3}$, we have w.p. $1 - \delta$,

$$
E_{W_2, W_2^*} A_2 \geq n^2 K^2 E_x (E_{W_1} \sigma^2(W_{1,1,:} x_i))^2 - (\log \frac{6}{\delta} \sigma_{max}^4 \sqrt{K}) K n^2
$$

$$
+ n^2 (\log \frac{6}{\delta})^2 K \sigma_{max}^4 - \log \frac{6}{\delta} \sqrt{8n} K \sigma_{max}^4
$$

$$
= \mathcal{O}(n^2 K^2).
$$

$\square$

**Lemma 11.** *Suppose* $W, W^*, x_i$ *are all i.i.d. random variables sampled from standard normal distribution. Then w.h.p,*

$$
E_{W_2, W_2^*} B_1 \leq \mathcal{O}(n^2 K^2).
$$

*Proof.* Expanding the expression of $B_2$, we have

$$
E_{W_2, W_2^*} B_1 = E_{W_2, W_2^*} \left( \sum_{p=1}^K \sum_{q=1}^d \left( \sum_{i=1}^n (\hat{y}_i - y_i) \frac{\partial \hat{y}_i}{\partial W_{1,p,q}} \right)^2 \right)
$$

$$
= E_{W_2, W_2^*} \left( \sum_{p=1}^K \sum_{q=1}^d \left( \sum_{i=1}^n (W_2 \sigma(W_1 x_i) - W_2^* \sigma(W_1^* x_i)) W_{2,1,p} \sigma'(W_{1,p:} x_i) x_{i,q} \right)^2 \right)
$$

$$
= \left( \sum_{p=1}^K \sum_{q=1}^d E_{W_2, W_2^*} \left( \sum_{i=1}^n (W_2 \sigma(W_1 x_i) - W_2^* \sigma(W_1^* x_i)) W_{2,1,p} \sigma'(W_{1,p:} x_i) x_{i,q} \right)^2 \right)
$$

$$
\leq 2 \left( \sum_{p=1}^K \sum_{q=1}^d E_{W_2, W_2^*} \left( \sum_{i=1}^n (W_2 \sigma(W_1 x_i)) W_{2,1,p} \sigma'(W_{1,p:} x_i) x_{i,q} \right)^2 \right)
$$

$$
+ 2 \left( \sum_{p=1}^K \sum_{q=1}^d E_{W_2, W_2^*} \left( \sum_{i=1}^n (W_2^* \sigma(W_1^* x_i)) W_{2,1,p} \sigma'(W_{1,p:} x_i) x_{i,q} \right)^2 \right)
$$

where the inequality is due to $(a + b)^2 \leq 2a^2 + 2b^2$. Expanding $W_2$, we have

$$
E_{W_2, W_2^*} \left( \sum_{i=1}^n (W_2 \sigma(W_1 x_i)) W_{2,1,p} \sigma'(W_{1,p:} x_i) x_{i,q} \right)^2
$$

$$
= E_{W_2, W_2^*} \left( \sum_{r=1}^K \sum_{i=1}^n (W_{2,1,r} \sigma(W_{1,r,:} x_i)) W_{2,1,p} \sigma'(W_{1,p:} x_i) x_{i,q} \right)^2
$$

$$
= \sum_{r=1}^K E_{W_2, W_2^*} \left( \sum_{i=1}^n (W_{2,1,r} \sigma(W_{1,r,:} x_i)) W_{2,1,p} \sigma'(W_{1,p:} x_i) x_{i,q} \right)^2
$$

$$
\leq 3 \sum_{r=1}^K \left( \sum_{i=1}^n (\sigma(W_{1,r,:} x_i)) \sigma'(W_{1,p:} x_i) x_{i,q} \right)^2,
$$

where the second equation is because $E(W_{2,1,r_1} W_{2,1,r_2} W_{2,1,p}^2) = 0$ as long as $r_1 \neq r_2$. The inequality is because $E(W_{2,1,r}^2 W_{2,1,p}^2) = 3$ if $r = p$ and $E(W_{2,1,r}^2 W_{2,1,p}^2) = 1 < 3$ if $r \neq q$. Similarly, we have

$$
E_{W_2, W_2^*} \left( \sum_{i=1}^n (W_2^* \sigma(W_1^* x_i)) W_{2,1,p} \sigma'(W_{1,p:} x_i) x_{i,q} \right)^2
$$

$$
= E_{W_2, W_2^*} \left( \sum_{r=1}^K \sum_{i=1}^n (W_{2,1,r}^* \sigma(W_{1,r,:}^* x_i)) W_{2,1,p} \sigma'(W_{1,p:} x_i) x_{i,q} \right)^2
$$

$$
= \sum_{r=1}^K E_{W_2, W_2^*} \left( \sum_{i=1}^n (W_{2,1,r}^* \sigma(W_{1,r,:}^* x_i)) W_{2,1,p} \sigma'(W_{1,p:} x_i) x_{i,q} \right)^2
$$

$$
= K \left( \sum_{i=1}^n \sigma(W_{1,r,:}^* x_i) \sigma'(W_{1,p:} x_i) x_{i,q} \right)^2.
$$

Therefore, we obtain

$$E_{W_2,W_2^*}B_1$$

$$\leq 2\sum_{p=1}^{K}\sum_{q=1}^{d}\left(3\sum_{r=1}^{K}\left(\sum_{i=1}^{n}\sigma(W_{1,r,:}x_i)\sigma'(W_{1,p:}x_i)x_{i,q}\right)^2 + K\left(\sum_{i=1}^{n}\sigma(W_{1,r,:}^*x_i)\sigma'(W_{1,p:}x_i)x_{i,q}\right)^2\right).$$

Since $\sigma(\cdot) \leq \sigma_{\max}$ and $\sigma(\cdot)' \leq \sigma'_{\max}$, we have

$$Pr(|\sigma(W_{1,r,:}x_i)\sigma'(W_{1,p:}x_i)x_{i,q}| \geq t) \leq Pr(|\sigma_{max}\sigma'_{max}x_{i,q}| \geq t)$$

$$\leq 2\exp(-\frac{t^2}{2\sigma_{max}^2\sigma_{max}'^2})$$

which implies $\sigma(W_{1,r,:}x_i)\sigma'(W_{1,p:}x_i)x_{i,q}$ is sub-exponential. Thus, we can apply Bernstein inequality to $\sum_{i=1}^{n}\sigma(W_{1,r,:}x_i)\sigma'(W_{1,p:}x_i)x_{i,q}$, to obtain w.p. $1-\delta$,

$$|\sum_{i=1}^{n}\sigma(W_{1,r,:}x_i)\sigma'(W_{1,p:}x_i)x_{i,q} - nE_{x_i}\sum_{i=1}^{n}\sigma(W_{1,r,:}x_i)\sigma'(W_{1,p:}x_i)x_{i,q}|$$

$$\leq \sqrt{\frac{n}{2}\log\frac{2}{\delta}}\sigma_{max}\sigma'_{max} + \sigma_{max}\sigma'_{max}\log\frac{2}{\delta}$$

and similarly w.p. $(1-\delta)$,

$$|\sum_{i=1}^{n}\sigma(W_{1,r,:}^*x_i)\sigma'(W_{1,p:}x_i)x_{i,q} - nE_{x_i}\sum_{i=1}^{n}\sigma(W_{1,r,:}^*x_i)\sigma'(W_{1,p:}x_i)x_{i,q}|$$

$$\leq \sqrt{\frac{n}{2}\log\frac{2}{\delta}}\sigma_{max}\sigma'_{max} + \sigma_{max}\sigma'_{max}\log\frac{2}{\delta}.$$

By union bound, we have w.p $1-2\delta$, the above two are both true. Plug them in the expression of $B_1$ and use $(a+b)^2 \leq 2a^2 + 2b^2$. Finally, we have w.p. $1-2\delta$,

$$E_{W_2,W_2^*}B_1$$

$$\leq 2\sum_{p=1}^{K}\sum_{q=1}^{d}\left(3\sum_{r=1}^{K}\left(\sum_{i=1}^{n}\sigma(W_{1,r,:}x_i)\sigma'(W_{1,p:}x_i)x_{i,q}\right)^2 + K\left(\sum_{i=1}^{n}\sigma(W_{1,r,:}^*x_i)\sigma'(W_{1,p:}x_i)x_{i,q}\right)^2\right)$$

$$\leq 2\sum_{p=1}^{K}\sum_{q=1}^{d}\left(3\sum_{r=1}^{K}\left(nE_{x_i}\sigma(W_{1,r,:}x_i)\sigma'(W_{1,p:}x_i)x_{i,q} + a_1\right)^2\right.$$

$$+ K\left(nE_{x_i}\sigma(W_{1,r,:}^*x_i)\sigma'(W_{1,p:}x_i)x_{i,q} + a_1\right)^2\Bigg)$$

$$\leq 4n^2\sum_{p=1}^{K}\sum_{q=1}^{d}\left(3\sum_{r=1}^{K}\left(E_{x_i}\sigma(W_{1,r,:}x_i)\sigma'(W_{1,p:}x_i)x_{i,q}\right)^2\right.$$

$$+ K\left(E_{x_i}\sigma(W_{1,r,:}^*x_i)\sigma'(W_{1,p:}x_i)x_{i,q}\right)^2 + 8Ka_1^2\Bigg),$$

where $a_1 = \sqrt{\frac{1}{2n}\log\frac{2}{\delta}}\sigma_{max}\sigma'_{max} + \frac{1}{n}\sigma_{max}\sigma'_{max}\log\frac{2}{\delta}$ is the extra error term. Note that this term is $O(\sqrt{\frac{1}{n}})$ and typically can be ignored. Now let us consider $E_{x_i}\sigma(W_{1,r,:}x_i)\sigma'(W_{1,p:}x_i)x_{i,q}$. Abuse the notation a little bit, let $X = W_{1,r,:}x_i, Y = W_{1,p:}x_i, Z = x_{i,q}$. Given $W_1$, they are all normal distribution, and the correlation is

$$\rho_{XZ} = \frac{W_{1,r,q}}{\sum_{b=1}^{d}W_{1,r,b}^2}$$

$$\rho_{YZ} = \frac{W_{1,p,q}}{\sum_{b=1}^{d}W_{1,p,b}^2}$$

$$\rho_{XY} = \frac{\sum_{b=1}^{d}W_{1,p,d}W_{1,r,d}}{\sqrt{\sum_{b=1}^{d}W_{1,p,b}^2\sum_{b=1}^{d}W_{1,r,b}^2}}$$

$$\sigma_x = \sqrt{\sum_{b=1}^{d}W_{1,r,b}^2}.$$

Apply lemma 6, we can get,

$$|E(\sigma(X)\sigma'(Y)Z)|$$
$$\leq (1-\rho_{XY}^2)^{-1}|\rho_{XZ} - \rho_{YZ}\rho_{XY}| \times \sigma_{max}\sigma'_{max}\sigma_x + (1-\rho_{XY}^2)^{-1}|-\rho_{XZ}\rho_{XY} + \rho_{YZ}| \times \sigma_{max}\alpha_G.$$

Note that by Chernoff bound and Lemma 7, w.p. $1 - 4\delta$,

$$|\rho_{XZ}| \leq \frac{2\log\frac{1}{\delta}}{d - \sqrt{d}\log\frac{1}{\delta}}$$

$$|\rho_{YZ}| \leq \frac{2\log\frac{1}{\delta}}{d - \sqrt{d}\log\frac{1}{\delta}}$$

$$|\rho_{XY}| \leq \sqrt{d}\frac{2\log\frac{1}{\delta}}{d - \sqrt{d}\log\frac{1}{\delta}}$$

$$\sigma_x \leq \sqrt{d + \sqrt{d}\log\frac{1}{\delta}}.$$

Plug them in the above inequality,we have w.p. $1 - 4\delta$,

$$E_{x_i}\sigma(W_{1,r,:}x_i)\sigma'(W_{1,p:}x_i)x_{i,q} = |E(\sigma(X)\sigma'(Y)Z)| \leq O(\frac{1}{\sqrt{d}})\sigma_{max}\sigma'_{max} + O(\frac{1}{d})\sigma_{max}\alpha_G.$$

Similarly we can get w.p. $1 - 4\delta$,

$$E_{x_i}\sigma(W_{1,r,:}^*x_i)\sigma'(W_{1,p:}x_i)x_{i,q} \leq O(\frac{1}{\sqrt{d}})\sigma_{max}\sigma'_{max} + O(\frac{1}{d})\sigma_{max}\alpha_G.$$

Thus, we have w.p. $1 - 10\delta$,

$$E_{W_2,W_2^*}B_1 \leq 4n^2 \sum_{p=1}^{K}\sum_{q=1}^{d}\left(3\sum_{r=1}^{K}\left(E_{x_i}\sigma(W_{1,r,:}x_i)\sigma'(W_{1,p:}x_i)x_{i,q}\right)^2\right.$$

$$+ K\left(E_{x_i}\sigma(W_{1,r,:}^*x_i)\sigma'(W_{1,p:}x_i)x_{i,q}\right)^2\Bigg)$$

$$\leq 4n^2\sum_{p=1}^{K}\sum_{q=1}^{d}\left(3K\left(\mathcal{O}(\frac{1}{\sqrt{d}}\sigma_{max}\sigma'_{max})\right)^2 + K\left(\mathcal{O}(\frac{1}{\sqrt{d}}\sigma_{max}\sigma'_{max})\right)^2\right)$$

$$= \mathcal{O}(K^2n^2),$$

which completes the proof. $\qquad\square$

**Lemma 12.** *Suppose $W, W^*, x_i$ are all i.i.d. random variables sampled from standard normal distribution. Then w.h.p,*

$$E_{W_2,W_2^*}B_2 \leq \mathcal{O}\left(n^2K^2\left(\frac{1}{d} + \frac{1}{\sqrt{d}} + \frac{1}{K}\right)\right).$$

*Proof.* Expanding the expression of $B_2$, we have

$$E_{W_2,W_2^*}B_2 = E_{W_2,W_2^*}\left(\sum_{q=1}^{K}\left(\sum_{i=1}^{n}(\hat{y}_i - y_i)\frac{\partial\hat{y}_i}{\partial W_{2,1,q}}\right)^2\right)$$

$$= E_{W_2,W_2^*}\left(\sum_{q=1}^{K}\left(\sum_{i=1}^{n}(W_2\sigma(W_1x_i) - W_2^*\sigma(W_1^*x_i))\sigma(W_{1,q,:}x_i)\right)^2\right)$$

$$= \sum_{q=1}^{K}\sum_{r=1}^{K}\left(\left(\sum_{i=1}^{n}\sigma(W_{1,r,:}x_i)\sigma(W_{1,q,:}x_i)\right)^2 + \left(\sum_{i=1}^{n}\sigma(W_{1,r,:}^*x_i)\sigma(W_{1,q,:}x_i)\right)^2\right),$$

where the third equation is because $W_2, W_{2*}$ are independent. Applying Hoeffding bound to $\sum_{i=1}^{n}\sigma(W_{1,r,:}x_i)\sigma(W_{1,q,:}x_i)$, we have w.p. $(1-\delta)$,

$$|\sum_{i=1}^{n}\sigma(W_{1,r,:}x_i)\sigma(W_{1,q,:}x_i) - nE_{x_i}\sigma(W_{1,r,:}x_i)\sigma(W_{1,q,:}x_i)| \leq \sqrt{\frac{n}{2}}\sigma_{max}^2\log\frac{2}{\delta}$$

and similarly w.p. $(1 - \delta)$,

$$|\sum_{i=1}^{n} \sigma(W_{1,r,:}^{*} x_i)\sigma(W_{1,q,:} x_i) - nE_{x_i}\sigma(W_{1,r,:}^{*} x_i)\sigma(W_{1,q,:} x_i)| \leq \sqrt{\frac{n}{2}}\sigma_{max}^2 \log\frac{2}{\delta}$$

By union bound, we have w.p $1 - 2\delta$, the above two are both true. Plug them in the expression of $B_2$, we have w.p. $1 - 2\delta$,

$$E_{W_2,W_2^{*}} B_2 = \sum_{q=1}^{K}\sum_{r=1}^{K}\left(\left(\sum_{i=1}^{n}\sigma(W_{1,r,:} x_i)\sigma(W_{1,q,:} x_i)\right)^2 + \left(\sum_{i=1}^{n}\sigma(W_{1,r,:}^{*} x_i)\sigma(W_{1,q,:} x_i)\right)^2\right)$$

$$\leq 2\sum_{q=1}^{K}\sum_{r=1}^{K}\left(n^2\left(E_{x_i}\sigma(W_{1,r,:} x_i)\sigma(W_{1,q,:} x_i)\right)^2 + n^2\left(E_{x_i}\sigma(W_{1,r,:}^{*} x_i)\sigma(W_{1,q,:} x_i)\right)^2 + n\sigma_{max}^4 \log^2\frac{2}{\delta}\right)$$

$$= 2n^2\sum_{q=1}^{K}\sum_{r=1}^{K}\left((E_{x_i}\sigma(W_{1,r,:} x_i)\sigma(W_{1,q,:} x_i))^2 + \left(E_{x_i}\sigma(W_{1,r,:}^{*} x_i)\sigma(W_{1,q,:} x_i)\right)^2\right)$$

$$+ 2K^2 n\sigma_{max}^4 \log^2\frac{2}{\delta}$$

$$= 2n^2\sum_{q=1}^{K}\sum_{r\neq q}\left((E_{x_i}\sigma(W_{1,r,:} x_i)\sigma(W_{1,q,:} x_i))^2 + \left(E_{x_i}\sigma(W_{1,r,:}^{*} x_i)\sigma(W_{1,q,:} x_i)\right)^2\right)$$

$$+ 2n^2\sum_{q=1}^{K}\left(\left(E_{x_i}\sigma^2(W_{1,q,:} x_i)\right)^2 + \left(E_{x_i}\sigma^2(W_{1,q,:}^{*} x_i)\right)^2\right)$$

$$+ 2K^2 n\sigma_{max}^4 \log^2\frac{2}{\delta},$$

where the first inequality is due to $(a + b)^2 \leq 2(a^2 + b^2)$.

For the first term, define

$$\rho_{q,r} = \frac{E_{x_i}(W_{1,r,:,} x_i W_{1,q,:,} x_i)}{E_{x_i}(W_{1,r,:,} x_i)^2 E_{x_i}(W_{1,q,:,} x_i)^2} = \frac{\sum_{y=1}^{d} W_{1,r,y} W_{1,q,y}}{\sqrt{\sum_{y=1}^{d} W_{1,r,y}^2}\sqrt{\sum_{y=1}^{d} W_{1,q,y}^2}}.$$

Apply lemma 8 to each $(E_{x_i}\sigma(W_{1,r,:} x_i)\sigma(W_{1,q,:} x_i))^2$. We have

$$(E_{x_i}\sigma(W_{1,r,:} x_i)\sigma(W_{1,q,:} x_i))^2$$

$$\leq \left(\left(\frac{\sigma_{\max} + 2\sqrt{\rho_{q,r}}\sigma_{\max} + 4\alpha_G\sqrt{1 - \rho_{q,r}}}{\sqrt{1 - \rho_{q,r}^2}}\right)\sigma_{\max}\sqrt{\rho_{q,r}}\right)^2$$

$$= \left(\frac{\sigma_{\max} + 2\sqrt{\rho_{q,r}}\sigma_{\max} + 4\alpha_G\sqrt{1 - \rho_{q,r}}}{\sqrt{1 - \rho_{q,r}^2}}\right)^2 \sigma_{\max}^2\rho_{q,r}$$

$$\leq 3\left(\frac{\sigma_{\max}^2 + 4\rho_{q,r}\sigma_{\max}^2 + 16\alpha_G^2(1 - \rho_{q,r})}{\sqrt{1 - \rho_{q,r}^2}}\right)\sigma_{\max}^2\rho_{q,r}$$

where the last inequality is due to $3a^2 + 3b^2 + 3c^2 \geq (a + b + c)^2$. By Lemma 7, we have w.p. $(1 - 3\delta)$,

$$\rho_{q,r} \leq \frac{\sqrt{d}\log\frac{2}{\delta}}{d - \sqrt{d}\log\frac{2}{\delta}}.$$

Therefore, plug in this value into the above inequality, we have w.p. $1 - 3\delta$,

$$(E_{x_i}\sigma(W_{1,r,:} x_i)\sigma(W_{1,q,:} x_i))^2$$

$$\leq 3\left(\frac{\sigma_{\max}^2 + 4\rho_{q,r}\sigma_{\max}^2 + 16\alpha_G^2(1 - \rho_{q,r})}{\sqrt{1 - \rho_{q,r}^2}}\right)\sigma_{\max}^2\rho_{q,r}$$

$$\simeq 3\left(\frac{\sigma_{\max}^2 + 4\frac{\sqrt{d}\log\frac{2}{\delta}}{d}\sigma_{\max}^2 + 16\alpha_G^2(1 - \frac{\sqrt{d}\log\frac{2}{\delta}}{d})}{\sqrt{1 - (\frac{\sqrt{d}\log\frac{2}{\delta}}{d})^2}}\right)\sigma_{\max}^2\frac{\sqrt{d}\log\frac{2}{\delta}}{d}$$

$$= \mathcal{O}(\frac{1}{\sqrt{d}} + \frac{1}{d})\log\frac{2}{\delta}.$$

Apply this for all $(q, r)$ pairs, and then use union bound, we have w.p. $1 - \delta$, for all $q \neq r$,

$$(E_{x_i}\sigma(W_{1,r,:}x_i)\sigma(W_{1,q,:}x_i))^2 \leq \mathcal{O}(\frac{1}{\sqrt{d}} + \frac{1}{d}) \log \frac{2K(K-1)}{\delta}$$

and thus

$$2n^2 \sum_{q=1}^{K} \sum_{r \neq q} \left( (E_{x_i}\sigma(W_{1,r,:}x_i)\sigma(W_{1,q,:}x_i))^2 + (E_{x_i}\sigma(W_{1,r,:}^*x_i)\sigma(W_{1,q,:}x_i))^2 \right)$$

$$\leq \mathcal{O}(n^2 K^2 (\frac{1}{\sqrt{d}} + \frac{1}{d}) \log \frac{2K(K-1)}{\delta})$$

$$\simeq \mathcal{O}(n^2 K^2 (\frac{1}{\sqrt{d}} + \frac{1}{d}) \log \frac{1}{\delta}).$$

For the second term, noting that $\sigma() \leq \sigma_{\max}$, we have

$$2n^2 \sum_{q=1}^{K} \left( (E_{x_i}\sigma^2(W_{1,q,:}x_i))^2 + (E_{x_i}\sigma^2(W_{1,q,:}^*x_i))^2 \right) \leq 2n^2 \sum_{q=1}^{K}(2\sigma_{\max}^2) = \mathcal{O}(n^2 K).$$

Combing all those terms, we have w.p. $1 - \delta$,

$$E_{W_2, W_2^*} B_2 \leq 2n^2 \sum_{q=1}^{K} \sum_{r \neq q} \left( (E_{x_i}\sigma(W_{1,r,:}x_i)\sigma(W_{1,q,:}x_i))^2 + (E_{x_i}\sigma(W_{1,r,:}^*x_i)\sigma(W_{1,q,:}x_i))^2 \right)$$

$$+ 2n^2 \sum_{q=1}^{K} \left( (E_{x_i}\sigma^2(W_{1,q,:}x_i))^2 + (E_{x_i}\sigma^2(W_{1,q,:}^*x_i))^2 \right)$$

$$+ 2K^2 n\sigma_{max}^4 \log^2 \frac{2}{\delta}$$

$$\leq \mathcal{O}(\frac{1}{\sqrt{d}} + \frac{1}{d}) \log \frac{2}{\delta} + \mathcal{O}(n^2 K) + \mathcal{O}(nK^2) \simeq \mathcal{O}(n^2 K^2 (\frac{1}{\sqrt{d}} + \frac{1}{d}) + n^2 K).$$

This completes the proof. $\qquad\qquad\qquad\qquad\qquad\qquad\qquad\qquad\qquad\qquad\qquad\qquad\square$

Now we are ready to prove Theorem 2.

*Proof.* Note that $f_i = (\hat{y}_i - y_i)$, we can apply the chain rule to have

$$n \sum_{i=1}^{n} ||\nabla f_i||_2^2 = n \sum_{i=1}^{n} \left( \sum_{p=1}^{K} \sum_{q=1}^{d} \left( \frac{\partial f_i}{\partial W_{1,p,q}} \right)^2 + \sum_{q=1}^{K} \left( \frac{\partial f_i}{\partial W_{2,1,q}} \right)^2 \right)$$

$$= n \sum_{i=1}^{n} (\hat{y}_i - y_i)^2 \left( \sum_{p=1}^{K} \sum_{q=1}^{d} \left( \frac{\partial \hat{y}_i}{\partial W_{1,p,q}} \right)^2 + \sum_{q=1}^{K} \left( \frac{\partial \hat{y}_i}{\partial W_{2,1,q}} \right)^2 \right)$$

$$= A_1 + A_2,$$

and

$$||\sum_{i=1}^{n} \nabla f_i||_2^2 = \left( \sum_{p=1}^{K} \sum_{q=1}^{d} \left( \sum_{i=1}^{n} \frac{\partial f_i}{\partial W_{1,p,q}} \right)^2 + \sum_{q=1}^{K} \left( \sum_{i=1}^{n} \frac{\partial f_i}{\partial W_{2,1,q}} \right)^2 \right)$$

$$= \left( \sum_{p=1}^{K} \sum_{q=1}^{d} \left( \sum_{i=1}^{n} (\hat{y}_i - y_i) \frac{\partial \hat{y}_i}{\partial W_{1,p,q}} \right)^2 + \sum_{q=1}^{K} \left( \sum_{i=1}^{n} (\hat{y}_i - y_i) \frac{\partial \hat{y}_i}{\partial W_{2,1,q}} \right)^2 \right)$$

$$= B_1 + B_2.$$

The goal is now to understand the behavior of $\frac{E_{W_2, W_2^*} A_1 + A_2}{E_{W_2, W_2^*} B_1 + B_2}$.

By Lemma 9, w.h.p,

$$E_{W_2, W_2^*} A_1 \geq \mathcal{O}(n^2 K^2 d).$$

By Lemma 10, we have w.h.p,

$$E_{W_2, W_2^*} A_2 \geq \mathcal{O}(n^2 K^2).$$

By Lemma 11, we have w.h.p,

$$E_{W_2, W_2^*} B_1 \leq \mathcal{O}(n^2 K^2).$$

By Lemma 12, we have w.h.p,

$$E_{W_2, W_2^*} B_2 \leq \mathcal{O}\left(n^2 K^2 \left(\frac{1}{d} + \frac{1}{\sqrt{d}} + \frac{1}{K}\right)\right).$$

Combing these four results we directly obtain the desired theorem. □

# C  Proofs for Multilayer Linear Neural Networks

We first present the main theorem for multilayer NNs.

**Theorem 6.** *Consider a LNN with $L \geq 2$ layers. Let the weight values $W_{l,p,q}$ for $l \in \{1, \ldots, L\}$ and $\mathbf{x}_i$ be independently drawn random variables from $\mathcal{N}(0,1)$. Let*

$$M = n^2 \prod_{\ell=0}^{L-1} K_\ell (K_\ell + 2)$$

*Then:*

$$\mathbb{E}[n \sum_{i=1}^{n} ||\nabla f_i||^2] = M \cdot L \left(1 + \prod_{\ell=1}^{L-1} \frac{K_\ell}{K_\ell + 2}\right),$$

$$\mathbb{E}[\sum_{i=1, j\neq i}^{n} \langle \nabla f_i, \nabla f_j \rangle] = M \cdot \frac{n-1}{n} \left(\sum_{\phi=0}^{L-1} \frac{L-\phi}{K_\phi - 1} \prod_{\ell=0}^{\phi} \frac{K_\ell - 1}{K_\ell + 2} + \frac{L}{K_0} \prod_{\ell=0}^{L-1} \frac{K_\ell}{K_\ell + 2}\right).$$

Given Theorem 6, Theorem 3, the main theorem for multilayer NNs, becomes a direct corollary.

Next, we prove Theorem 6. We start by stating a few general lemmas that will be necessary in the proof.

## C.1  Models, Assumptions and Notations

Let us denote $W = \prod_{\ell=1}^{L} W_\ell$ and $W^* = \prod_{\ell=1}^{L} W_\ell^*$. We will also need the following notation:

$$r_{a,p} = \left(\prod_{\ell=a+2}^{L} W_\ell\right) W_{a+1,:,p} \qquad l_{a,q} = W_{a-1,q,:} \left(\prod_{\ell=1}^{a-2} W_\ell\right) \qquad l_{a,q,s}^b = W_{a-1,q,:} \left(\prod_{\ell=b}^{a-2} W_\ell\right) W_{b-1,:,s},$$

where

$$\left(\prod_{\ell=a+2}^{L} W_\ell\right) W_{a+1,:,p} = \begin{cases} W_{L,:,p} & \text{if } a = L-1 \\ 1 & \text{if } a = L \end{cases} \qquad W_{a-1,q,:} \left(\prod_{\ell=1}^{a-2} W_\ell\right) = \begin{cases} e_q^T & \text{if } a = 1 \\ W_{1,q,:} & \text{if } a = 2 \end{cases},$$

where $e_q \in R^d$ is the $q$-th unit vector in the $d$ dimensional space.

Then we can write the differential as follows:

$$\frac{\partial f_i}{\partial W_{a,p,q}} = (\hat{y}_i - y_i) \left(\prod_{\ell=a+2}^{L} W_\ell\right) W_{a+1,:,p} W_{a-1,q,:} \left(\prod_{\ell=1}^{a-2} W_\ell\right) x_i = (W x_i - W^* x_i) \, r_{a,p}(l_{a,q} x_i).$$

Furthermore, let $W_{(a+2):L} = \prod_{\ell=a+2}^{L} W_\ell$.

By default, we define $\prod_{i=k}^{n} a_i = 1$ if $k > n$.

## C.2  Some Helper Lemmas

We would need the Isserlis Theorem **(author?)** [49]. The following lemma can derived from the Isserlis Theorem.

**Lemma 13.** *Let* $x \in \mathbb{R}^d$ *such that* $x_i$ *is i.i.d.* $\sim \mathcal{N}(0,1)$. *Then*

$$E_x \left(a^\mathsf{T} x\right)^2 = ||a||_2^2$$

$$E_x \left(a^\mathsf{T} x\right)^4 = 3||a||_2^4$$

$$E_x \left(a^\mathsf{T} x\right)^8 = 105||a||_2^8$$

$$E_x \left(a^\mathsf{T} x b^\mathsf{T} x\right) = a \cdot b$$

$$E_x \left(a^\mathsf{T} x\right)^2 \left(b^\mathsf{T} x\right)^2 = 2||a \cdot b||^2 + ||a||_2^2 ||b||^2$$

$$E_x \left(a^\mathsf{T} x\right)^4 \left(b^\mathsf{T} x\right)^4 \leq 105||a||_2^4 ||b||^4 \leq 105 \left(E_x \left(a^\mathsf{T} x\right)^2 \left(b^\mathsf{T} x\right)^2\right)^2.$$

**Lemma 14.** *Let* $B = (b_1, b_2, \cdots, b_{d_a}) \in \mathbb{R}^{d_c \times d_a}$ *be a random matrix whose elements are all i.i.d* $N(0,1)$, *and* $c = (c_1, c_2, \cdots, b_{d_c}) \in \mathbb{R}^{d_c}$ *be a constant. Let* $a_i = c^\mathsf{T} b_i$. *Then we have*

$$E \left(\sum_{i=1}^{d_a} a_i^2\right)^2 = d_a (d_a + 2) \left(\sum_{j=1}^{d_c} c_j^2\right)^2. \tag{C.1}$$

*Proof.* By linearity of Expectation, we have

$$\begin{aligned} E \left(\sum_{i=1}^{d_a} a_i^2\right)^2 &= E \left(\sum_{i=1}^{d_a} \sum_{j=1, j \neq i}^{d_a} a_i^2 a_j^2\right) + E \left(\sum_{i=1}^{d_a} a_i^4\right) \\ &= \sum_{i=1}^{d_a} \sum_{j=1, j \neq i}^{d_a} E \left((c^\mathsf{T} b_i)^2 (c^\mathsf{T} b_j)^2\right) + \sum_{i=1}^{d_a} E \left((c^\mathsf{T} b_i)^4\right) \\ &= \sum_{i=1}^{d_a} \sum_{j=1, j \neq i}^{d_a} \left(\sum_{k=1}^{d_c} c_k^2\right)^2 + \sum_{i=1}^{d_a} 3 \left(\sum_{k=1}^{d_c} c_k^2\right)^2 \\ &= d_a (d_a + 2) \left(\sum_{j=1}^{d_c} c_j^2\right)^2, \end{aligned} \tag{C.2}$$

where the third equation is due to Lemma 13. $\qquad\square$

**Lemma 15.** *Let* $G = (g_1; g_2; \cdots; g_{d_g}) \in \mathbb{R}^{d_g \times d_a}$ *be a random matrix whose elements are all i.i.d* $N(0,1)$, $x_s, x_t \in \mathbb{R}^{d_x}$ *be i.i.d* $N(0,1)$, *and* $A \in \mathbb{R}^{d_a \times d_x}$ *be a constant. Then we have*

$$E \left(\sum_{j=1}^{d_g} g_j A x_s g_j A x_t\right)^2 = \sum_{i=1}^{d_g} \sum_{j=1}^{d_g} E \left(\sum_{u=1}^{d_x} g_j A_{:,u} g_i A_{:,u}\right)^2. \tag{C.3}$$

*Proof.* By linearity of expectation, we have

$$\begin{aligned} E \left(\sum_{j=1}^{d_g} g_j A x_s g_j A x_t\right)^2 &= \sum_{i=1}^{d_g} \sum_{j=1}^{d_g} E \left(g_j A x_s g_j A x_t g_i A x_s g_i A x_t\right) \\ &= \sum_{i=1}^{d_g} \sum_{j=1}^{d_g} E \left(\sum_{u=1}^{d_x} g_j A_{:,u} g_i A_{:,u} \sum_{v=1}^{d_x} g_j A_{:,v} g_i A_{:,v}\right) \\ &= \sum_{i=1}^{d_g} \sum_{j=1}^{d_g} E \left(\sum_{u=1}^{d_x} g_j A_{:,u} g_i A_{:,u}\right)^2, \end{aligned} \tag{C.4}$$

where the second equality is taking expectation over $x$. $\qquad\square$

**Lemma 16.** *if all elements in* $W$ *are i.i.d standard normal distribution, we have*

$$E \left(\sum_{s=1}^{K_a} r_{a,s}^2 r_{a,p}^2\right) = (K_a + 2) F_a = (K_a + 2) \left(\prod_{\ell=a+1}^{L-1} K_\ell (K_\ell + 2)\right).$$

*Proof.*

$$E_{W_{a+1}} \left( \sum_{s=1}^{K_a} r_{a,s}^2 r_{a,p}^2 \right)$$

$$= E_{W_{a+1}} \left( \sum_{s=1}^{K_a} \left( W_{(a+2):L} W_{a+1,:,s} \right)^2 \left( W_{(a+2):L} W_{a+1,:,p} \right)^2 \right)$$

$$= E_{W_{a+1}} \left( \sum_{s=1,s\neq p}^{K_a} \left( W_{(a+2):L} W_{a+1,:,s} \right)^2 \left( W_{(a+2):L} W_{a+1,:,p} \right)^2 + \left( W_{(a+2):L} W_{a+1,:,p} \right)^4 \right)$$

$$= (K_a + 2) \left( E_{W_{a+1}} \left( \left( W_{(a+2):L} W_{a+1,:,p} \right)^2 \right) \right)^2$$

$$= (K_a + 2) \left( \sum_{r=1}^{K_{a+1}} \left( W_{(a+3):L} W_{a+2,:,r} \right)^2 \right)^2.$$

The first equation expands the expression of the original formula. The second equation split the summation over $s$ into two parts, the case when $p = s$ and the case $p \neq s$. The third equation uses the fact that $E\|a^T x\|_2^4 = 3\|a\|_2^4$ from Lemma 13, and that all $W_{a+1,:,s}$ are symmetric and thus the expectation of the sum is essentially $K_a - 1$ times the expectation of each value.

By Lemma 14, we have

$$E_{W_{a+2}} \left( \left( \sum_{r=1}^{K_{a+1}} \left( W_{(a+3):L} W_{a+2,:,r} \right)^2 \right)^2 \right) = K_{a+1} (K_{a+1} + 2) \left( \sum_{r=1}^{K_{a+2}} \left( W_{a+4:L} W_{a+3,:,r} \right)^2 \right)^2.$$

Note that now the formula on the right side has the same form of that on the left side. This actually means that we can use induction over $a$ to further simplify it. Formally, let

$$F_a = \left( \sum_{r=1}^{K_{a+1}} \left( W_{(a+3):L} W_{a+2,:,r} \right)^2 \right)^2.$$

Then the above equation becomes for all $a \leq L - 4$,

$$E_{W_{a+2}} (F_a) = K_{a+1} (K_{a+1} + 2) F_{a+1},$$

which implies

$$E(F_a) = K_{a+1} (K_{a+1} + 2) E(F_{a+1}).$$

Now we prove that by induction,

$$E(F_a) = \left( \prod_{\ell=a+1}^{L-1} K_\ell (K_\ell + 2) \right), \forall a \leq L - 3.$$

When $a = L - 3$, we have

$$E\left(F_{L-3}\right) = E\left(\sum_{r=1}^{K_{L-2}} \left(W_L W_{L-1,:,r}\right)^2\right)^2$$

$$= E\left(\sum_{r=1}^{K_{L-2}} \sum_{v=1}^{K_{L-2}} \left(W_L W_{L-1,:,r}\right)^2 \left(W_L W_{L-1,:,v}\right)^2\right)$$

$$= E\left(\sum_{r=1}^{K_{L-2}} \sum_{v=1,v\neq r}^{K_{L-2}} \left(W_L W_{L-1,:,r}\right)^2 \left(W_L W_{L-1,:,v}\right)^2 + \sum_{r=1}^{K_{L-2}} \left(W_L W_{L-1,:,r}\right)^4\right)$$

$$= \sum_{r=1}^{K_{L-2}} \sum_{v=1,v\neq r}^{K_{L-2}} E\left(W_L W_{L-1,:,r}\right)^2 \left(W_L W_{L-1,:,v}\right)^2 + \sum_{r=1}^{K_{L-2}} E\left(W_L W_{L-1,:,r}\right)^4$$

$$= \sum_{r=1}^{K_{L-2}} \sum_{v=1,v\neq r}^{K_{L-2}} E_{W_L} E_{W_{L-1}} \left(W_L W_{L-1,:,r}\right)^2 \left(W_L W_{L-1,:,v}\right)^2$$

$$+ \sum_{r=1}^{K_{L-2}} E_{W_L} E_{W_{L-1}} \left(W_L W_{L-1,:,r}\right)^4$$

$$= \sum_{r=1}^{K_{L-2}} \sum_{v=1,v\neq r}^{K_{L-2}} E_{W_L} \left(E_{W_{L-1}} \left(W_L W_{L-1,:,r}\right)^2\right)^2 + \sum_{r=1}^{K_{L-2}} E_{W_L} E_{W_{L-1}} \left(W_L W_{L-1,:,r}\right)^4$$

$$= \sum_{r=1}^{K_{L-2}} \sum_{v=1,v\neq r}^{K_{L-2}} E_{W_L} \left(\sum_{t=1}^{K_{L-1}} \left(W_{L,:,t}\right)^2\right)^2 + 3\sum_{r=1}^{K_{L-2}} E_{W_L} \left(\sum_{t=1}^{K_{L-1}} \left(W_{L,:,t}\right)^2\right)^2$$

$$= K_{L-2} \left(K_{L-2} + 2\right) E_{W_L} \left(\sum_{t=1}^{K_{L-1}} \left(W_{L,:,t}\right)^2\right)^2$$

$$= K_{L-2} \left(K_{L-2} + 2\right) K_{L-1} \left(K_{L-1} + 2\right).$$

The first and second equations expand the expression of $F_L$. The third equation splits the summation over $v$ into 2 parts, the case when $v = r$ and the case when $v \neq r$. The forth equation uses the linearity of expectation and the fifth equation uses conditional expectation. The sixth equation uses the fact that $W_{L-1}$ are all i.i.d standard normal distribution. The seventh equation uses the fact that $E\|a^T x\|_2^4 = 3\|a\|^4$ from Lemma 13. The last two equations are simple algebra.

Assume that when $a = \theta$,

$$E(F_\theta) = \left(\prod_{\ell=\theta+1}^{L-1} K_\ell \left(K_\ell + 2\right)\right).$$

When $a = \theta - 1$, we have

$$E\left(F_{\theta-1}\right) = K_\theta \left(K_\theta + 2\right) E\left(F_\theta\right)$$

$$= \left(\prod_{\ell=\theta}^{L-1} K_\ell \left(K_\ell + 2\right)\right).$$

Thus, by induction,

$$E(F_a) = \left(\prod_{\ell=a+1}^{L-1} K_\ell \left(K_\ell + 2\right)\right), \forall a \leq L - 3.$$

Therefore,

$$E\left(\sum_{s=1}^{K_a} r_{a,s}^2 r_{a,p}^2\right) = \left(K_a + 2\right) E(F_a) = \left(K_a + 2\right) \left(\prod_{\ell=a+1}^{L-1} K_\ell \left(K_\ell + 2\right)\right).$$

$\square$

**Lemma 17.** *If all elements in $W, W^*, x$ are i.i.d standard normal distribution, then*

$$E\left(\sum_{t=1}^{K_{a-1}} (l_{a,q}x_i)^2 (l_{a,t}x_i)^2\right) = (K_{a-1}+2)\left(\prod_{\ell=0}^{a-2} K_\ell (K_\ell+2)\right).$$

*Proof.* Similar to the proof of Lemma 16. □

**Lemma 18.** *If all elements in $W, W^*, x$ are i.i.d standard normal distribution, then*

$$E\left(\sum_{t=1}^{K_{a-1}} (l_{a,q,v})^2 (l_{a,t,v})^2\right) = (K_{a-1}+2)\left(\prod_{\ell=1}^{a-2} K_\ell (K_\ell+2)\right).$$

*Proof.* Similar to the proof of Lemma 16. □

**Lemma 19.**

$$E\left(\sum_{s,t=1}^{K_{b-2}} \left(\sum_{v=1}^{K_{a-1}} l_{a,v,s}^b l_{a,v,t}^b\right)^2\right) = \left(\prod_{\ell=b-2}^{a-1} K_\ell (K_\ell-1)\right)\left(\sum_{\phi=b-2}^{a-1} \frac{1}{K_\phi-1} \prod_{\ell=\phi+1}^{a-1} \frac{K_\ell+2}{K_\ell-1}\right),$$

*and in particular,*

$$E\left(\sum_{s,t=1}^{K_0} \left(\sum_{v=1}^{K_{a-1}} l_{a,v,s}^0 l_{a,v,t}^0\right)^2\right) = \left(\prod_{\ell=0}^{a-1} K_\ell (K_\ell-1)\right)\left(\sum_{\phi=0}^{a-1} \frac{1}{K_\phi-1} \prod_{\ell=\phi+1}^{a-1} \frac{K_\ell+2}{K_\ell-1}\right).$$

*Proof.* We will prove the result using recurrent formula. Let us first note that

$$E\left(\sum_{s,t=1}^{K_{b-2}} \left(\sum_{v=1}^{K_{a-1}} l_{a,v,s}^b l_{a,v,t}^b\right)^2\right)$$

$$= K_{b-2}E\left(\left(\sum_{v=1}^{K_{a-1}} \left(l_{a,v,s}^b\right)^2\right)^2\right) + K_{b-2}(K_{b-2}-1) E\left(\left(\sum_{v=1}^{K_{a-1}} l_{a,v,s}^b l_{a,v,t}^b\right)^2\right)$$

$$= K_{b-2} \prod_{\ell=b-1}^{a-1} K_\ell (K_\ell+2) + K_{b-2}(K_{b-2}-1) E\left(\left(\sum_{v=1}^{K_{a-1}} l_{a,v,s}^b l_{a,v,t}^b\right)^2\right),$$

where the first equation splits the summation over $s$ into two cases, the case when $s=t$ and the case when $s \neq t$. Note that $l_{a,v,s}^b$ are symmetric over all $s$, and thus the summation in the first case becomes $K_{b-2}$ times a single term. Similarly, in the second term, we have $K_{b-2}K_{b-2}-1$ as the coefficient. The second equation essentially plugs in the value of the $E\left(\left(\sum_{v=1}^{K_{a-1}} l_{a,v,s}^b\right)^2\right)$, which is a sum of squares and we already know how to compute it using lemma 14.

Now let us turn to the term $E\left(\left(\sum_{v=1}^{K_{a-1}} l_{a,v,s}^b l_{a,v,t}^b\right)^2\right)$. Let us further split $W_b$ in this is term, and we obtain

$$E\left(\left(\sum_{v=1}^{K_{a-1}} l_{a,v,s}^b l_{a,v,t}^b\right)^2\right) = E\left(\sum_{v_1=1}^{K_{a-1}}\sum_{v_2=1}^{K_{a-1}} l_{a,v_1,s}^b l_{a,v_1,t}^b l_{a,v_2,s}^b l_{a,v_2,t}^b\right)$$

$$= E\left(\sum_{v_1=1}^{K_{a-1}}\sum_{v_2=1}^{K_{a-1}} l_{a,v_1,:}^{b+1} W_{b,:,s} l_{a,v_1,:}^{b+1} W_{b,:,t} l_{a,v_2,:}^{b+1} W_{b,:,s} l_{a,v_2,:}^{b+1} W_{b,:,t}\right)$$

$$= E\left(\sum_{v_1=1}^{K_{a-1}}\sum_{v_2=1}^{K_{a-1}}\sum_{s=1}^{K_{b-1}}\sum_{t=1}^{K_{b-1}} l_{a,v_1,s}^{b+1} l_{a,v_1,t}^{b+1} l_{a,v_2,s}^{b+1} l_{a,v_2,t}^{b+1}\right)$$

$$= E\left(\sum_{s,t=1}^{K_{b-1}} \left(\sum_{v=1}^{K_{a-1}} l_{a,v,s}^{b+1} l_{a,v,t}^{b+1}\right)^2\right),$$

where the first equation is simply expanding the square of summations, the second equation is splitting the expression of $\ell_{a,v,s}^b$, the third equation is computing the expectation over $W_b$, and the final equation is changing the order of summation. Combining the above two main equations, we effectively obtain the following equation.

$$
E\left(\sum_{s,t=1}^{K_{b-2}}\left(\sum_{v=1}^{K_{a-1}} l_{a,v,s}^b l_{a,v,t}^b\right)^2\right)
$$

$$
= K_{b-2}\prod_{\ell=b-1}^{a-1} K_\ell\left(K_\ell+2\right) + K_{b-2}\left(K_{b-2}-1\right) E\left(\sum_{s,t=1}^{K_{b-1}}\left(\sum_{v=1}^{K_{a-1}} l_{a,v,s}^{b+1} l_{a,v,t}^{b+1}\right)^2\right).
$$

This holds for every $b \leq a - 1$. Now Let us define $f(b) = E\left(\sum_{s,t=1}^{K_{b-2}}\left(\sum_{v=1}^{K_{a-1}} l_{a,v,s}^b l_{a,v,t}^b\right)^2\right)/\prod_{\ell=b-2}^{a-1} K_\ell\left(K_\ell-1\right)$. The above equation now becomes

$$
f(b) = \frac{1}{K_{b-2}-1}\prod_{\ell=b-1}^{a-1}\frac{K_\ell+2}{K_\ell-1} + f(b+1).
$$

One can easily check that $f(a+1) = \frac{1}{K_{a-1}-1}$ and that $f(a) = \frac{K_{a-1}+2}{K_{a-1}-1}\frac{1}{K_{a-2}-1} + \frac{1}{K_{a-1}-1}$. Therefore, by induction, we can easily obtain

$$
f(b) = \frac{1}{K_{b-2}-1}\prod_{\ell=b-1}^{a-1}\frac{K_\ell+2}{K_\ell-1} + f(b+1)
$$

$$
= \cdots
$$

$$
= \sum_{\phi=b-2}^{a-1}\frac{1}{K_\phi-1}\prod_{\ell=\phi+1}^{a-1}\frac{K_\ell+2}{K_\ell-1},
$$

where we use the notation $\prod_{u=i}^j = 1, i > j$ for simplicity. By plugging in back $f(b)$ to the expression of the expectation, we have

$$
E\left(\sum_{s,t=1}^{K_{b-2}}\left(\sum_{v=1}^{K_{a-1}} l_{a,v,s}^b l_{a,v,t}^b\right)^2\right) = \left(\prod_{\ell=b-2}^{a-1} K_\ell\left(K_\ell-1\right)\right)\left(\sum_{\phi=b-2}^{a-1}\frac{1}{K_\phi-1}\prod_{\ell=\phi+1}^{a-1}\frac{K_\ell+2}{K_\ell-1}\right).
$$

This completes the proof. $\qquad\square$

## C.3 Computing the Expectation

**Theorem 7.** *If $\forall a, W_{a,p,q}^*, W_{a,p,q}, x_i$ are all i.i.d $\sim \mathcal{N}(0,1)$, then*

$$
E(\|\frac{\partial f_i}{\partial W_{a,p,q}}\|^2) = \frac{K_0\left(K_0+2\right)}{K_a K_{a-1}}\left(\prod_{\ell=1}^{L-1} K_\ell\left(K_\ell+2\right) + \prod_{\ell=1}^{L-1} K_\ell^2\right).
$$

*Proof.* We start by writing the expectation as follows.

$$
E\left(\|\frac{\partial f_i}{\partial W_{a,p,q}}\|\right)^2 = E\left((Wx_i - W^* x_i)r_{a,p}(l_{a,q}x_i)\right)^2
$$

$$
= E\left((Wx_i)^2 r_{a,p}^2\left(l_{a,q}x_i\right)^2\right) + E\left(\left(W^* x_i\right)^2 r_{a,p}^2\left(l_{a,q}x_i\right)^2\right),
$$

where we plug in the expression of the derivative in the first equation. The second equation uses the fact that $W$ and $W^*$ are 0-means independent random variables.

For the first term, computing the expectation over $W_a$, we have

$$
E_{W_a}\left((Wx_i)^2 r_{a,p}^2\left(l_{a,q}x_i\right)^2\right) = \sum_{s=1}^{K_a}\sum_{t=1}^{K_{a-1}} r_{a,s}^2 r_{a,p}^2\left(l_{a,q}x_i\right)^2\left(l_{a,t}x_i\right)^2
$$

$$
= \sum_{s=1}^{K_a} r_{a,s}^2 r_{a,p}^2\sum_{t=1}^{K_{a-1}}\left(l_{a,q}x_i\right)^2\left(l_{a,t}x_i\right)^2,
$$

where the first equation uses the fact that $W_a$ only appears in $W$ where $W = W_{(a+1):L}W_aW_{(1):(a-1)}$, and all elements in $W_a$ are i.i.d 0-means. Note that $r_{i,j}$ and $l_{k,\ell}$ are independent, so we can compute their expectation separately. By Lemma 16, we have

$$E\left(\sum_{s=1}^{K_a} r_{a,s}^2 r_{a,p}^2\right) = (K_a + 2) F_a = (K_a + 2)\left(\prod_{\ell=a+1}^{L-1} K_\ell (K_\ell + 2)\right).$$

By Lemma 17, we have

$$E\left(\sum_{t=1}^{K_{a-1}} (l_{a,q}x_i)^2 (l_{a,t}x_i)^2\right) = (K_{a-1} + 2)\left(\prod_{\ell=0}^{a-2} K_\ell (K_\ell + 2)\right).$$

Combining those two equations we have

$$E\left((Wx_i)^2 r_{a,p}^2 (l_{a,q}x_i)^2\right) = \left(\prod_{\ell=0, \ell \notin \{a,a-1\}}^{L-1} K_\ell (K_\ell + 2)\right)(K_{a-1} + 2)(K_a + 2).$$

For the second term, note that $(W^*x_i)^2$, $r_{a,p}^2$ and $(l_{a,q}x_i)^2$ are independent given $x_i$. Thus, we can compute the conditional expectation separately.

$$E_{W^*}\left((W^*x_i)^2\right) = E_{W^*}\left(\left(\prod_{t=1}^{L} W_t^*x_i\right)^2\right)$$

$$= E_{W^*}\left(\left(W_L^* \prod_{t=1}^{L-1} W_t^*x_i\right)^2\right)$$

$$= E_{W^*}\left(\sum_{\alpha=1}^{K_{L-1}} W_{L,:,\alpha}^{*,2}\left(W_{L-1,\alpha,:}^* \prod_{t=1}^{L-2} W_t^*x_i\right)^2\right)$$

$$= K_{L-1}E_{W^*}\left(\left(W_{L-1,\alpha,:}^* \prod_{t=1}^{L-2} W_t^*x_i\right)^2\right)$$

$$= \cdots$$

$$= \prod_{\ell=1}^{L-1} K_\ell \sum_{k=1}^{K_0} x_{i,k}^2,$$

where the first two equations are simply plugging in the expression of $W^*$. The third equation uses the fact that $W_a^*$ are i.i.d. 0-mean. The fourth equation uses the fact that $W_{L,:,a}$ are symmetric. The fifth equation uses induction to finally obtain the last equation. Similarly,

$$E_{l_{a,q}}\left((l_{a,q}x_i)^2\right) = \prod_{\ell=1}^{a-2} K_\ell \sum_{k=1}^{K_0} x_{i,k}^2,$$

and

$$E_{r_{a,p}}\left((r_{a,p})^2\right) = \prod_{\ell=a+1}^{L-1} K_\ell.$$

Hence, the second term becomes

$$E\left((W^*x_i)^2 r_{a,p}^2 (l_{a,q}x_i)^2\right) = E_{x_i}\left(E_{W^*}(W^*x_i)^2 E_{r_{a,p}}\left(r_{a,p}^2\right) E_{l_{a,q}}(l_{a,q}x_i)^2\right)$$

$$= E_{x_i}\left(\prod_{\ell=1}^{L-1} K_\ell \sum_{k=1}^{K_0} x_{i,k}^2 \prod_{\ell=1}^{a-2} K_\ell \sum_{k=1}^{K_0} x_{i,k}^2 \prod_{\ell=a+1}^{L-1} K_\ell\right)$$

$$= \frac{1}{K_{a-1}K_a}\prod_{\ell=1}^{L-1} K_\ell^2 E_{x_i}\left(\left(\sum_{k=1}^{K_0} x_{i,k}^2\right)^2\right)$$

$$= \frac{1}{K_{a-1}K_a}\prod_{\ell=0}^{L-1} K_\ell^2.$$

Combing both terms finishes the proof. $\qquad\square$

**Theorem 8.** *If $\forall \ell, p, q, i, W^*_{\ell,p,q}, W_{\ell,p,q}, x_i$ are all i.i.d $\sim \mathcal{N}(0,1)$, then we have*

$$E\left(||\nabla f_i||^2\right) = L\left(K_0\left(K_0+2\right)\left(\prod_{\ell=1}^{L-1} K_\ell\left(K_\ell+2\right) + \prod_{\ell=1}^{L-1} K_\ell^2\right)\right)$$

*Proof.* This can be directly obtained from the last theorem by summing over $a, p, q$. $\square$

Remarks: One can verify that when $L = 2$, this reduces to the 2-layer case and we have $E(||\frac{\partial f_i}{\partial W_{a,p,q}}||^2) = 2d(d+2)(2K+2)K$, which agrees with the 2-layer analysis.

**Theorem 9.** *If $W_{\ell,p,q}, W^*_{\ell,p,q}, x_i, x_j, i \neq j$ are all i.i.d $\sim \mathcal{N}(0,1)$, then we have*

$$E\left(\frac{\partial f_i}{\partial W_{a,p,q}}\right)\left(\frac{\partial f_j}{\partial W_{a,p,q}}\right)$$
$$= \frac{1}{K_a K_{a-1}}\left(\prod_{\ell=0}^{L-1} K_\ell\left(K_\ell+2\right)\right)\left(\sum_{\phi=0}^{a-1}\frac{1}{K_\phi-1}\prod_{\ell=0}^{\phi}\frac{K_\ell-1}{K_\ell+2} + \frac{1}{K_0}\prod_{\ell=0}^{L-1}\frac{K_\ell}{K_\ell+2}\right),$$

*Proof.* Note that

$$E\left(\frac{\partial f_i}{\partial W_{a,p,q}}\right)\left(\frac{\partial f_j}{\partial W_{a,p,q}}\right) = E\left((Wx_i - W^*x_i)(Wx_j - W^*x_j)r_{a,p}^2(l_{a,q}x_i)(l_{a,q}x_j)\right)$$
$$= E\left((W^*x_i)(W^*x_j)r_{a,p}^2(l_{a,q}x_i)(l_{a,q}x_j)\right)$$
$$+ E\left((Wx_i)(Wx_j)r_{a,p}^2(l_{a,q}x_i)(l_{a,q}x_j)\right),$$

where we plug in the expression of the derivative into the first equation, and the second equation uses the fact that $E(WW^*) = 0$ since $W, W^*$ are independent i.i.d. random variables.

For the first term, we have

$$E\left((W^*x_i)(W^*x_h)r_{a,p}^2(l_{a,q}x_i)(l_{a,q}x_h)\right)$$
$$= E\left(r_{a,p}^2\sum_{s=1}^{K_0}\sum_{t=1}^{K_0} W^*_{2:L}W^*_{1,:,s}W^*_{2:L}W^*_{1,:,t}W_{a-1,q,:}W_{2:a-2}W_{1,:,s}W_{a-1,q,:}W_{2:a-2}W_{1,:,t}\right)$$
$$= E\left(r_{a,p}^2\sum_{s=1}^{K_0}\left(W^*_{2:L}W^*_{1,:,s}\right)^2\left(W_{a-1,q,:}W_{2:a-2}W_{1,:,s}\right)^2\right),$$

where the first equation is because of taking expectation over $x$ and $x_i, x_j$ are i.i.d 0-mean, while the second equation is because we take the expectation over $W_1$ where again $W_1$ are independent and 0-mean.

Since $r, W, W^*$ are independent, we have

$$E\left(r_{a,p}^2\sum_{s=1}^{K_0}\left(W^*_{2:L}W^*_{1,:,s}\right)^2\left(W_{a-1,q,:}W_{2:a-2}W_{1,:,s}\right)^2\right)$$
$$= E\left(r_{a,p}^2\right)\sum_{s=1}^{K_0} E\left(W^*_{2:L}W^*_{1,:,s}\right)^2 E\left(W_{a-1,q,:}W_{2:a-2}W_{1,:,s}\right)^2.$$

Applying the fact that $E_x(||a^T x||_2^2) = ||a||_2^2$ from Lemma 13, we have

$$E\left(r_{a,p}^2\right) = E\left(\left(\left(\prod_{\ell=a+2}^{L} W_\ell\right) W_{a+1,:,p}\right)^2\right)$$

$$= E\left(\left\|\prod_{\ell=a+2}^{L} W_\ell\right\|^2\right)$$

$$= E\left(\sum_{v=1}^{K_{a+1}} \left(\prod_{\ell=a+3}^{L} W_\ell W_{a+2,:,v}\right)^2\right)$$

$$= K_{a+1} E\left(\left(\prod_{\ell=a+3}^{L} W_\ell W_{a+2,:,v}\right)^2\right)$$

$$= K_{a+1} E(r_{a+1,v}^2)$$

$$= K_{a+1} K_{a+2} E(r_{a+2,p}^2)$$

$$= \cdots$$

$$= \prod_{\ell=a+1}^{L-1} K_\ell.$$

Similarly, we have

$$E\left(W_{2:L}^* W_{1,:,s}^*\right)^2 = \prod_{\ell=1}^{L-1} K_\ell$$

and

$$E\left(W_{a-1,q,:} W_{2:a-2} W_{1,:,s}\right)^2 = \prod_{\ell=1}^{a-2} K_\ell.$$

Hence,

$$E\left(r_{a,p}^2 \sum_{s=1}^{K_0} \left(W_{2:L}^* W_{1,:,s}^*\right)^2 \left(W_{a-1,q,:} W_{2:a-2} W_{1,:,s}\right)^2\right) = K_0 \prod_{\ell=1}^{L-1} K_\ell^2 \cdot \frac{1}{K_{a-1} K_a}.$$

For the second term, we have

$$E\left((W x_i)(W x_j) r_{a,p}^2 (l_{a,q} x_i)(l_{a,q} x_j)\right)$$

$$= E\left(\sum_{s=1}^{K_0} \sum_{t=1}^{K_0} r_{a,p}^2 W_{2:L} W_{1,:,s} W_{2:L} W_{1,:,t} l_{a,q,s} l_{a,q,t}\right)$$

$$= E\left(\sum_{s,t=1}^{K_0} \sum_{u=1}^{K_a} \sum_{v=1}^{K_{a-1}} r_{a,p}^2 r_{a,u}^2 l_{a,v,s} l_{a,v,t} l_{a,q,s} l_{a,q,t}\right),$$

where we use similar tricks as in the first term, i.e., the first equation is due to taking expectation over $x$, and the last equation is by taking expectation over $W_a$. Note that $r$ and $l$ are independent, we can compute their expectation separately. For computation convenience, let us now take into account of summation over $p, q$ as well. This is essentially compute the sum of the derivative over $W_a$ instead of $W_{a,p,q}$. By Lemma 16,

$$\sum_p E\left(\sum_{u=1}^{K_a} r_{a,p}^2 r_{a,u}^2\right) = K_a (K_a + 2) \prod_{\ell=a+1}^{L-1} K_\ell (K_\ell + 2) = \prod_{\ell=a}^{L-1} K_\ell (K_\ell + 2),$$

which implies

$$E\left(\sum_{u=1}^{K_a} r_{a,p}^2 r_{a,u}^2\right) = \frac{1}{K_a} \prod_{\ell=a}^{L-1} K_\ell (K_\ell + 2).$$

Now let us consider $l$.

$$\sum_{q=1}^{K_{a-1}} E\left(\sum_{s,t=1}^{K_0} \sum_{v=1}^{K_{a-1}} l_{a,v,s} l_{a,v,t} l_{a,q,s} l_{a,q,t}\right) = E\left(\sum_{s,t=1}^{K_0} \left(\sum_{v=1}^{K_{a-1}} l_{a,v,s} l_{a,v,t}\right)^2\right).$$

By Lemma 19, we have

$$E\left(\sum_{s,t=1}^{K_0}\left(\sum_{v=1}^{K_{a-1}} l_{a,v,s}^0 l_{a,v,t}^0\right)^2\right) = \left(\prod_{\ell=0}^{a-1} K_\ell\left(K_\ell - 1\right)\right)\left(\sum_{\phi=0}^{a-1}\frac{1}{K_\phi - 1}\prod_{\ell=\phi+1}^{a-1}\frac{K_\ell + 2}{K_\ell - 1}\right),$$

which implies

$$E\left(\sum_{s,t=1}^{K_0}\sum_{v=1}^{K_{a-1}} l_{a,v,s} l_{a,v,t} l_{a,q,s} l_{a,q,t}\right) = \frac{1}{K_{a-1}}\left(\prod_{\ell=0}^{a-1} K_\ell\left(K_\ell - 1\right)\right)\left(\sum_{\phi=0}^{a-1}\frac{1}{K_\phi - 1}\prod_{\ell=\phi+1}^{a-1}\frac{K_\ell + 2}{K_\ell - 1}\right),$$

Combing those two terms, we have

$$E\left((Wx_i)(Wx_j) r_{a,p}^2 (l_{a,q}x_i)(l_{a,q}x_j)\right)$$

$$=E\left(\sum_{s,t=1}^{K_0}\sum_{u=1}^{K_a}\sum_{v=1}^{K_{a-1}} r_{a,p}^2 r_{a,u}^2 l_{a,v,s} l_{a,v,t} l_{a,q,s} l_{a,q,t}\right)$$

$$= \left(\frac{1}{K_a}\prod_{\ell=a}^{L-1} K_\ell\left(K_\ell + 2\right)\right)\left(\frac{1}{K_{a-1}}\left(\prod_{\ell=0}^{a-1} K_\ell\left(K_\ell - 1\right)\right)\left(\sum_{\phi=0}^{a-1}\frac{1}{K_\phi - 1}\prod_{\ell=\phi+1}^{a-1}\frac{K_\ell + 2}{K_\ell - 1}\right)\right)$$

$$= \left(\frac{1}{K_a K_{a-1}}\prod_{\ell=0}^{L-1} K_\ell\left(K_\ell + 2\right)\right)\left(\left(\prod_{\ell=0}^{a-1}\frac{K_\ell - 1}{K_\ell + 2}\right)\left(\sum_{\phi=0}^{a-1}\frac{1}{K_\phi - 1}\prod_{\ell=\phi+1}^{a-1}\frac{K_\ell + 2}{K_\ell - 1}\right)\right)$$

$$= \left(\frac{1}{K_a K_{a-1}}\prod_{\ell=0}^{L-1} K_\ell\left(K_\ell + 2\right)\right)\left(\sum_{\phi=0}^{a-1}\frac{1}{K_\phi - 1}\prod_{\ell=0}^{\phi}\frac{K_\ell - 1}{K_\ell + 2}\right).$$

Summing the two terms from the original expression, we finally have

$$E\left(\frac{\partial f_i}{\partial W_{a,p,q}}\right)\left(\frac{\partial f_j}{\partial W_{a,p,q}}\right)$$

$$= E\left((W^*x_i)(W^*x_j) r_{a,p}^2 (l_{a,q}x_i)(l_{a,q}x_j)\right)$$

$$+ E\left((Wx_i)(Wx_j) r_{a,p}^2 (l_{a,q}x_i)(l_{a,q}x_j)\right)$$

$$= K_0\prod_{\ell=1}^{L-1} K_\ell^2 \cdot \frac{1}{K_{a-1}K_a} + \left(\frac{1}{K_a K_{a-1}}\prod_{\ell=0}^{L-1} K_\ell\left(K_\ell + 2\right)\right)\left(\sum_{\phi=0}^{a-1}\frac{1}{K_\phi - 1}\prod_{\ell=0}^{\phi}\frac{K_\ell - 1}{K_\ell + 2}\right)$$

$$= \frac{1}{K_a K_{a-1}}\left(\prod_{\ell=0}^{L-1} K_\ell\left(K_\ell + 2\right)\right)\left(\sum_{\phi=0}^{a-1}\frac{1}{K_\phi - 1}\prod_{\ell=0}^{\phi}\frac{K_\ell - 1}{K_\ell + 2} + \frac{1}{K_0}\prod_{\ell=0}^{L-1}\frac{K_\ell}{K_\ell + 2}\right),$$

which completes the proof. □

**Theorem 10.** *If* $W_{\ell,p,q}, x_i, x_j, i \neq j$ *are all i.i.d* $\sim \mathcal{N}(0,1)$, *then we have*

$$E\left(\langle\nabla f_i, \nabla f_j\rangle\right) = \left(\prod_{\ell=0}^{L-1} K_\ell\left(K_\ell + 2\right)\right)\left(\sum_{\phi=0}^{a-1}\frac{L-\phi}{K_\phi - 1}\prod_{\ell=0}^{\phi}\frac{K_\ell + 2}{K_\ell - 1} + \frac{L}{K_0}\prod_{\ell=0}^{L-1}\frac{K_\ell}{K_\ell + 2}\right).$$

*Proof.* From Theorem 9, we have

$$E\left(\frac{\partial f_i}{\partial W_{a,p,q}}\right)\left(\frac{\partial f_j}{\partial W_{a,p,q}}\right)$$

$$= \frac{1}{K_a K_{a-1}}\left(\prod_{\ell=0}^{L-1} K_\ell\left(K_\ell + 2\right)\right)\left(\sum_{\phi=0}^{a-1}\frac{1}{K_\phi - 1}\prod_{\ell=0}^{\phi}\frac{K_\ell - 1}{K_\ell + 2} + \frac{1}{K_0}\prod_{\ell=0}^{L-1}\frac{K_\ell}{K_\ell + 2}\right).$$

Summing over $p, q$, we have

$$\sum_{p=1}^{K_a}\sum_{q=1}^{K_{a-1}} E\left(\frac{\partial f_i}{\partial W_{a,p,q}}\right)\left(\frac{\partial f_j}{\partial W_{a,p,q}}\right)$$

$$= \left(\prod_{\ell=0}^{L-1} K_\ell\left(K_\ell + 2\right)\right)\left(\sum_{\phi=0}^{a-1}\frac{1}{K_\phi - 1}\prod_{\ell=0}^{\phi}\frac{K_\ell - 1}{K_\ell + 2} + \frac{1}{K_0}\prod_{\ell=0}^{L-1}\frac{K_\ell}{K_\ell + 2}\right).$$

Thus, we have

$$E\left(\langle\nabla f_i,\nabla f_j\rangle\right) = \sum_{a=1}^{L}\sum_{p=1}^{K_a}\sum_{q=1}^{K_{a-1}} E\left(\frac{\partial f_i}{\partial W_{a,p,q}}\right)\left(\frac{\partial f_j}{\partial W_{a,p,q}}\right)$$

$$= \sum_{a=1}^{L}\left(\prod_{\ell=0}^{L-1}K_\ell\left(K_\ell+2\right)\right)\left(\sum_{\phi=0}^{a-1}\frac{1}{K_\phi-1}\prod_{\ell=0}^{\phi}\frac{K_\ell-1}{K_\ell+2}+\frac{1}{K_0}\prod_{\ell=0}^{L-1}\frac{K_\ell}{K_\ell+2}\right)$$

$$= \left(\prod_{\ell=0}^{L-1}K_\ell\left(K_\ell+2\right)\right)\left(\sum_{a=1}^{L}\sum_{\phi=0}^{a-1}\frac{1}{K_\phi-1}\prod_{\ell=0}^{\phi}\frac{K_\ell-1}{K_\ell+2}+\frac{L}{K_0}\prod_{\ell=0}^{L-1}\frac{K_\ell}{K_\ell+2}\right)$$

$$= \left(\prod_{\ell=0}^{L-1}K_\ell\left(K_\ell+2\right)\right)\left(\sum_{\phi=0}^{a-1}\frac{L-\phi}{K_\phi-1}\prod_{\ell=0}^{\phi}\frac{K_\ell-1}{K_\ell+2}+\frac{L}{K_0}\prod_{\ell=0}^{L-1}\frac{K_\ell}{K_\ell+2}\right).$$

$\square$

Finally we arrive at the main theorem.

**Theorem 6.** *Consider a LNN with $L \geq 2$ layers. Let the weight values $W_{l,p,q}$ for $l \in \{1,\ldots,L\}$ and $\mathbf{x}_i$ be independently drawn random variables from $\mathcal{N}(0,1)$. Let*

$$M = n^2\prod_{\ell=0}^{L-1}K_\ell\left(K_\ell+2\right)$$

*Then:*

$$\mathbb{E}[n\sum_{i=1}^{n}||\nabla f_i||^2] = M\cdot L\left(1+\prod_{\ell=1}^{L-1}\frac{K_\ell}{K_\ell+2}\right),$$

$$\mathbb{E}[\sum_{i=1,j\neq i}^{n}\langle\nabla f_i,\nabla f_j\rangle] = M\cdot\frac{n-1}{n}\left(\sum_{\phi=0}^{L-1}\frac{L-\phi}{K_\phi-1}\prod_{\ell=0}^{\phi}\frac{K_\ell-1}{K_\ell+2}+\frac{L}{K_0}\prod_{\ell=0}^{L-1}\frac{K_\ell}{K_\ell+2}\right).$$

*Proof.* This can be directly achieved from Theorem 8 and Theorem 10. $\square$