[Reviews · NeurIPS 2018]

Reviewer 1



This paper studies the influence of depth and width on the performance of large-batch training of neural networks. By presenting some theoretical results on shallow non-linear networks and deep linear networks, they argue that larger width would allow for larger batch training, which is thus beneficial for distributed computation. Overall I find the paper well-written. Discussion and presentation of results are clear. Systematical experiments on synthetic and real-world datasets could demonstrate to some extent the selling points. I also like the main messages/observations in the paper. However I would like to understand a bit more the implication of results on practical training of neural networks. In particular, as mentioned in the paper, using larger batch sizes can lead to speedup gains in distributed settings, but in practice we often care more about generalization performance than just fitting the training set. At this point, I'm curious to see how's the generalization of tested networks as a function of width resp. batch size? It might be worth to mention that apart from the benefits of width already listed in section 2, prior work have also shown that larger width could lead to a well-behaved loss landscape which makes the training of these networks become possible at all (see e.g. optimization landscape and expressivity of deep cnns, 2018, and the references therein). Line 161: it seems that W^* needs to be introduced. -- Thanks for the clear rebuttal. I find the results and analysis of the paper interesting and a very good complement to prior theoretical work on the influence of network width on training of practical deep NNs, given the fact that most of previous work focus on the power of depth. I believe the paper is worth to be published at NIPS and hope to see further developments in this direction.

Reviewer 2



It has been wide discussed on how to develop algorithms allow large batches, so that one could train neural networks in a distributed environment. The paper investigates the effect of network width on the performance of large-batch training both theoretically and experimentally. The authors claim that with the same number of parameters, it is more likely to train neural networks using proper large batches easily with a wide network architecture. The theoretical support on 2-layers linear/nonlinear networks and multilayer linear networks is also given. The paper is well-written and easy to follow. It extends the analysis of Dong el.s' paper about gradient diversity and addresses partly the open question proposed in their works about neural networks. The experiments are well-designed to show the advantage of wide network in many aspects. Plenty of extensive demonstrations (multilayer non-linear or residual networks) are provided and the results are informative. I would therefore vote an acceptance for the paper. However, I have some concerns and I hope it could be addressed 1. There are plenty of unused hyper-parameters in Theorems (c_1, c_2, c_max, c_sup, W^*, W, etc.) and also it's vague to say "with arbitrary constant probability", "sufficient large" or "high probability" in formal stated theorems. If you do want these constants, could be better to have some dependence on it. 2. In the proofs of Theorem 1, probability \delta is used to derive the final bound of B_S(w), while I'm not sure, how could you derive the last equation over line 443? should \delta be bounded? please also show how you handle the negative \Theta item in the nominator. 3. The conclusion of Theorem 2 and 3 shows the ratio of the expectation, and it does not imply a direct bound for the batch threshold B_S(w), so please explain more about the intuition that supports your conjecturing, otherwise the theoretical conclusion might be over-claimed. 4. To measure the convergence performance among different networks architectures and batch-size, a fixed convergence threshold is pre-defined, e.g., MNIST to 96% accuracy. I'm wondering is the value sensitive to your claims? It usually takes more epochs to achieve better accuracy in the final phase of training, considering the probably better expression of deep nets, what would happen if you trying to train MNIST to 100% accuracy? 5. I maybe misunderstanding somewhere, but in figure 3, how could you have same parameters with say (b) K=21, L=1 and K=17, L=10? Based on the comments above, I would therefore vote a weak accept to the work, but I would like to appreciate it if you could make it a better work. minors: 1. Line 33, duplicated reference [18]; 2. Line 256, redundant word "this" 3. Missing clarification of order symbols (\Theta and \Omega) in Theorems;

Reviewer 3



Large mini-batch size can speed up the training of deep learning models using a lot of data. There are many researches in the direction. The paper studies how network structure (width and depth) affects the mini-batch size during training. It presents several interesting findings. It found that gradient diversity increases monotonically as width increase and depth decreases. Seem like wider networks provide more space for the gradients to become diverse, so wider networks allow larger batch size than deeper one. The results of experiments confirm theoretic findings. I was able to follow the paper and the topic is interesting. For fully connected and ResNet models, the results are inspiring. I think it will be more interesting that the study evolves other layers like filters, pooling, etc. We think depth helps build better models. This study show wide/shallow models actually trains faster for a given accuracy and parameter budget. In Figure 2 (b), why does batch size drop sharply? also for (c), why does batch size increase exponentially?